# A cleaved METTL3 potentiates the METTL3–WTAP interaction and breast cancer progression

Chaojun Yan[1†], Jingjing Xiong[1†], Zirui Zhou[1†], Qifang Li[1†], Chuan Gao[1], Mengyao Zhang[1], Liya Yu[1], Jinpeng Li[2], Ming-Ming Hu[3], Chen-Song Zhang[4], Cheguo Cai[1], Haojian Zhang[3], Jing Zhang[1]*

[1]Department of Thyroid and Breast Surgery, Medical Research Institute, Frontier Science Center for Immunology and Metabolism, Zhongnan Hospital of Wuhan University, Wuhan University, Wuhan, China; [2]Department of Thyroid and Breast Surgery, Zhongnan Hospital of Wuhan University, Wuhan, China; [3]Frontier Science Center for Immunology and Metabolism, Medical Research Institute, Wuhan University, Wuhan, China; [4]State Key Laboratory for Cellular Stress Biology, Innovation Center for Cell Signaling Network School of Life Sciences, Xiamen University, Fujian, China

*For correspondence:
Jing_zhang@whu.edu.cn

[†]These authors contributed equally to this work

Competing interest: The authors declare that no competing interests exist.

**Abstract** $N^6$-methyladenosine (m$^6$A) methylation of RNA by the methyltransferase complex (MTC), with core components including METTL3–METTL14 heterodimers and *Wilms' tumor 1-associated protein* (WTAP), contributes to breast tumorigenesis, but the underlying regulatory mechanisms remain elusive. Here, we identify a novel cleaved form METTL3a (residues 239–580 of METTL3). We find that METTL3a is required for the METTL3–WTAP interaction, RNA m$^6$A deposition, as well as cancer cell proliferation. Mechanistically, we find that METTL3a is essential for the METTL3–METTL3 interaction, which is a prerequisite step for recruitment of WTAP in MTC. Analysis of m$^6$A sequencing data shows that depletion of METTL3a globally disrupts m$^6$A deposition, and METTL3a mediates mammalian target of rapamycin (mTOR) activation via m$^6$A-mediated suppression of TMEM127 expression. Moreover, we find that METTL3 cleavage is mediated by proteasome in an mTOR-dependent manner, revealing positive regulatory feedback between METTL3a and mTOR signaling. Our findings reveal METTL3a as an important component of MTC, and suggest the METTL3a–mTOR axis as a potential therapeutic target for breast cancer.

## eLife assessment

This study presents the **valuable** finding that a cleaved form of METTL3 (termed METTL3a) has an essential role in regulating the assembly of the METTL3-METTL14-WTAP complex. The evidence supporting the claims of the authors is **solid**, and the work will be of interest to medical biologists working on breast cancer.

## Introduction

$N^6$-methyladenosine (m$^6$A) methylation is the most abundant and evolutionarily conserved internal mRNA modification in eukaryotes, and is involved in numerous aspects of mRNA metabolism including RNA splicing, localization, stability, and translation (*Huang et al., 2020*; *Meyer et al., 2015*; *Li et al., 2017a*; *Lin et al., 2016*; *Xiao et al., 2016*). m$^6$A mRNA modification participates in numerous physiological processes, such as neurogenesis (*Li et al., 2017c*; *Livneh et al., 2020*; *Wang et al., 2018*; *Yoon*

*et al., 2017*), embryo development (*Batista et al., 2014*; *Li et al., 2018*; *Mendel et al., 2018*; *Wang et al., 2014*), and reproductive system development (*Haussmann et al., 2016*; *Ivanova et al., 2017*; *Kasowitz et al., 2018*; *Xia et al., 2018*; *Zheng et al., 2013*). Dysregulation of m6A deposition is associated with various diseases including cancer pathogenesis and drug resistance (*Huang et al., 2020*; *Lan et al., 2021*; *Zhang et al., 2021*), such as the core m6A methyltransferase (METTL3) promotes the progression of breast cancer through elevating expression of mammalian hepatitis BX-interacting protein and inhibiting tumor suppressor *let-7g* (*Cai et al., 2018*), inhibits tumor immune surveillance by upregulating PD-L1 mRNA level (*Wan et al., 2022*), and contributes to breast cancer tamoxifen resistance by upregulation of AK4 expression (*Liu et al., 2020*).

The m6A mRNA modification is deposited by a methyltransferase complex (MTC), the METTL3–METTL14–WTAP complex serves as the core of MTC, METTL3 is the core catalytic component, METTL14 contributes to substrate RNA recognition, triggering METTL3 methylation activity (*Huang et al., 2019*; *Wang et al., 2016a*; *Wang et al., 2016b*; *Wang et al., 2014*), and WTAP has been recognized as an essential adaptor responsible for recruitment of METTL3–METTL14 heterodimer to nuclear speckles. The m6A modifications are globally reduced in the absence of WTAP (*Liu et al., 2014*; *Ping et al., 2014*). The underlying regulatory mechanisms of m6A machinery have recently begun to emerge. In particular, ERK was found to phosphorylate METTL3 and WTAP, resulting in stabilization of

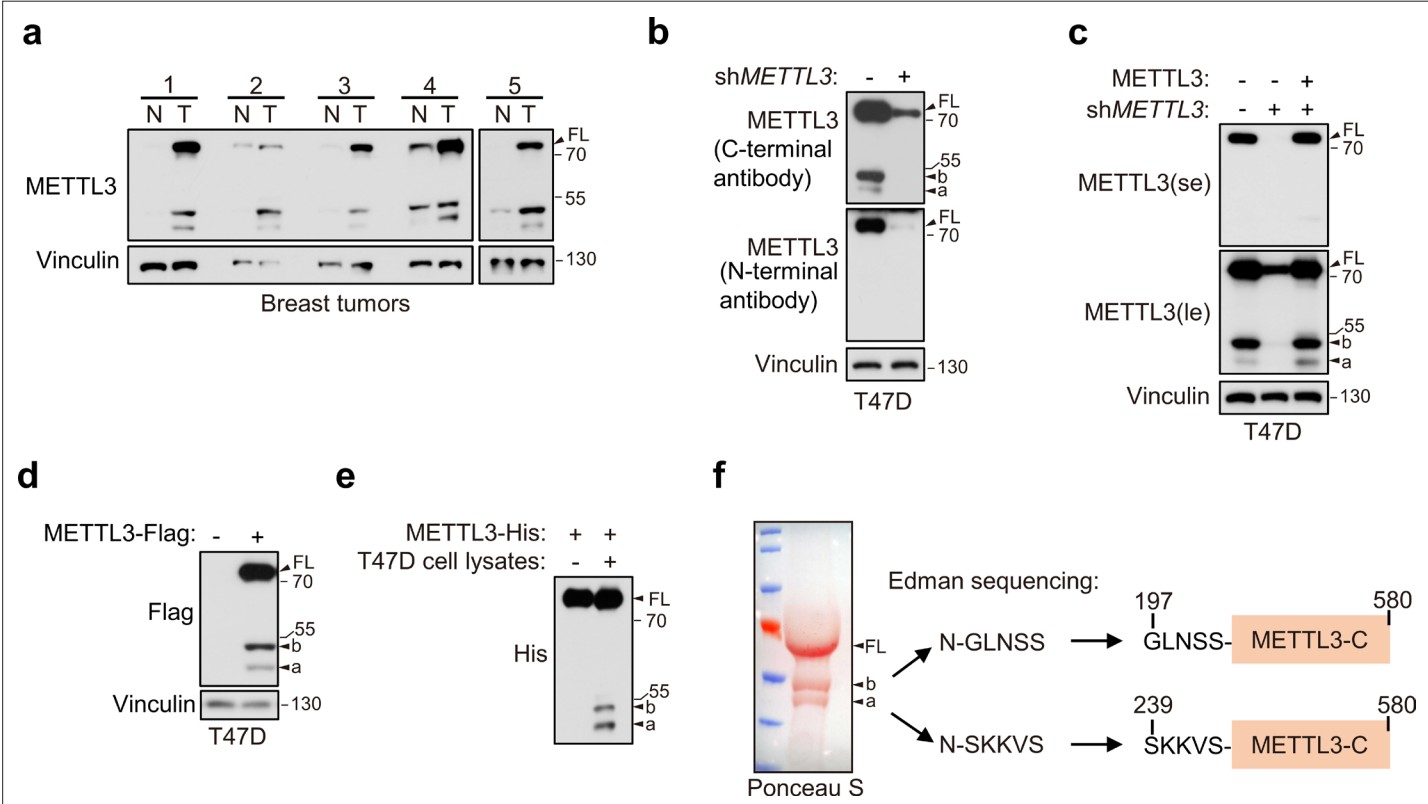

**Figure 1.** Identification of two novel C-terminal short forms of METTL3. (**a**) Immunoblot of lysates from paired breast cancer patient non-tumor (N) and tumor (T) tissues. (**b**) Immunoblot of T47D cells infected with lentivirus encoding *METTL3* shRNA or control shRNA by antibodies recognizing N- or C-terminal of METTL3 as indicated. (**c**) Immunoblot of T47D cells infected with retrovirus encoding sh*METTL3* resistant pMSCV-METTL3 without tag or control vector followed by *METTL3* shRNA or control shRNA infection. (**d**) Immunoblot of T47D cells infected with retrovirus encoding pMSCV-METTL3 with C-terminal Flag tag. (**e**) Purified recombinant METTL3-His protein was incubated with or without T47D cell lysates at 37°C for 1 hr followed by immunoblot with anti-His antibody. (**f**) Ponceau S staining and the N-terminal sequence of the short forms of METTL3 determined by Edman sequencing are shown. FL indicates the full-length of METTL3. The short forms are labeled as a and b.

The online version of this article includes the following source data and figure supplement(s) for figure 1:

**Source data 1.** Unedited western blot images for *Figure 1*.

**Figure supplement 1.** The short forms bands exist in lung cancer cells and other cell lines.

**Figure supplement 1—source data 1.** Unedited western blot images for *Figure 1—figure supplement 1*.

the m⁶A MTC (*Sun et al., 2020b*). Three other studies reported that mTORC1 promotes WTAP translation (*Cho et al., 2021*), S-adenosylmethionine (SAM) synthesis (*Villa et al., 2021*), and activation of chaperonin CCT complex (*Tang et al., 2021*), leading to the global increase in m⁶A modifications. In this study, we have identified a cleaved form of METTL3, METTL3a (residues 239–580), that is essential for the METTL3–WTAP interaction, RNA m⁶A methylation, and breast cancer progression.

## Results

### Identification of two novel C-terminal short forms METTL3a (residues 239–580) and METTL3b (residues 197–580)

METTL3 is well known to promote breast tumorigenesis (*Huang et al., 2020*). We obtained breast cancer patient tumor tissues and found that METTL3 was upregulated in tumor samples (*Figure 1a*). Notably, we observed two short bands, which were all elevated synchronously with the full-length (FL) of METTL3. To determine whether these bands were derived from METTL3, we constructed METTL3 knockdown in human T47D breast cancer cell lines by METTL3 shRNA and found that the amount of these two short bands was abolished (*Figure 1b*). Reconstitution of exogenous METTL3 by no tagged pMSCV-METTL3 construct in these cells rescued the expression levels of these two bands (*Figure 1c*). Interestingly, we observed that the two short bands were recognized by antibody against the C-terminal, but not N-terminal of METTL3 (*Figure 1b*), suggesting that these short forms are derived from the C-terminal of METTL3. Furthermore, we constructed T47D cell line stably expressing exogenous METTL3 by using C-terminal Flag-tagged pMSCV-METTL3 plasmid and also observed these short bands by Flag antibody (*Figure 1d*). These results indicate that these short bands are METTL3 C-terminal short forms, and we designated these two short forms as METTL3a and METTL3b as indicated in the figures. Further, we also observed these short bands in other cell lines, including lung cancer cell lines, renal cancer cell lines, HCT116, and MEF (mouse embryonic fibroblast) cell lines (*Figure 1—figure supplement 1*), indicating that these short forms may be ubiquitously expressed.

Since both endogenous and exogenous METTL3 expression in cells could generate METTL3a and METTL3b, we hypothesized that they are most likely generated from post-translational regulation. To test this possibility, we purified recombinant protein of METTL3 and incubated it with T47D cell lysates, and found that the purified METTL3-His could generate these short forms upon incubation with T47D cell lysates (*Figure 1e*). This result indicates that these short forms are derived from post-translational products of METTL3. To determine the specific METTL3 regions contained in these short forms, we separated each clean short form through sodium dodecyl sulfate–polyacrylamide gel electrophoresis (SDS–PAGE) and Ponceau S staining of the purified recombinant METTL3-Flag followed by edman sequencing of their respective first five N-terminal amino acids. The sequencing results showed that the METTL3a N-terminal sequence was ₂₃₉SKKVS₂₄₃, while that of METTL3b was ₁₉₇GLNSS₂₀₁, revealing that residues 239–580 and 197–580 comprise METTL3a and METTL3b, respectively (*Figure 1f*). We therefore identified two C-terminal short forms METTL3a (residues 239–580) and METTL3b (residues 197–580).

### METTL3a and METTL3b are post-translational products through highly conserved residues

We next generated point mutations or deletions in those conserved terminal residues, which further revealed that deletion of METTL3 Q238 (Δ238) or Q239 (Δ239) could most significantly reduce the expression of METTL3a in 293T cells (*Figure 2a, b*). Similarly, deletion of L198 (Δ198) or L199 (Δ199) resulted in the strongest suppression of METTL3b expression (*Figure 2c, d*). Notably, endogenous deletion of either L198 or Q238 (METTL3-Δ238 or METTL3-Δ198) by CRISPR/Cas9 knock-in (KI) could lead to decreased expression of both short forms (*Figure 2e, f*). Furthermore, we confirmed these post-translationally generated short forms by incubation of purified recombinant METTL3 with cell lysates of T47D, and found that METTL3-WT-His, but not METTL3-Δ238-His, METTL3-Δ198-His, or METTL3-(Δ198+Δ238)-His, could generate these short forms (*Figure 2g* and *Figure 2—figure supplement 2a*). Collectively, these results suggest that METTL3a and METTL3b are post-translationally generated through the highly conserved residues.

Through immunofluorescence assays, we found that METTL3a, METTL3b, METTL3-Δ198, and METTL3-Δ238 were mainly localized to nucleus, similar to METTL3 WT (*Figure 2—figure*

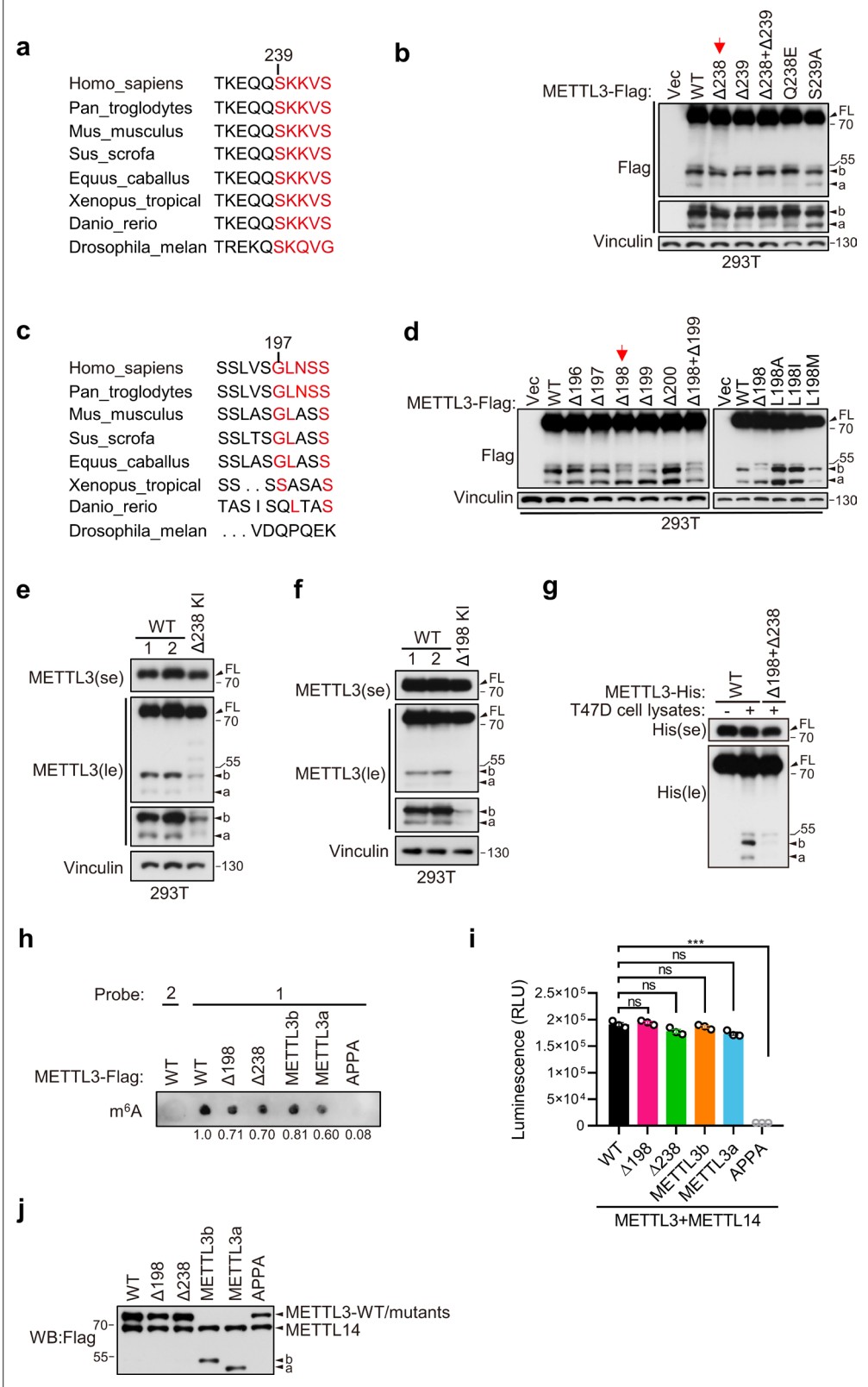

**Figure 2.** METTL3a (residues 239–580) and METTL3b (residues 197–580) are post-translational products through highly conserved residues. Sequence alignments of the conserved residues on (234–244) (**a**) and (192–201) (**c**) of METTL3. (**b**, **d**) Immunoblot of 293T cells transfected with empty vector (Vec) or METTL3-WT (wild-type) or the indicated mutants. Immunoblot analysis of 293T cells with CRISPR knock-in (KI) mediated deletion of Q238 (**e**)

*Figure 2 continued on next page*

*Figure 2 continued*

or L198 (**f**). (**g**) Purified recombinant METTL3-WT-His or METTL3-(Δ198+Δ238)-His protein were incubated with or without T47D cell lysates at 37°C for 1 hr followed by immunoblot with anti-His antibody. (**h–j**) The in vitro protein methylation activity was tested using purified METTL3-WT-Flag and its mutant proteins in combination with co-purified Flag-METTL14 and RNA-probe. The methylation activity was measured by using dot blot (**h**) or the Promega bioluminescence assay (**i**), and the immunoblot of those purified proteins are shown in (**j**). Error bars represent mean ± standard deviation (SD), unpaired *t*-test. ***p < 0.001, ns denotes no significance. FL indicates the full-length of METTL3. The short forms are labeled as a and b. se and le indicated short exposure and long exposure, respectively.

The online version of this article includes the following source data and figure supplement(s) for figure 2:

**Source data 1.** Unedited western blot images for *Figure 2*.

**Source data 2.** Table related to *Figure 2i*.

**Figure supplement 1.** Distribution of METTL3-WT and its mutants in breast cancer cells.

**Figure supplement 1—source data 1.** Images related to *Figure 2—figure supplement 1*.

**Figure supplement 2.** The short form METTL3c is dispensable for METTL3-mediated functions.

**Figure supplement 2—source data 1.** Unedited western blot images for *Figure 2—figure supplement 2a–d, f–h, and j–l*.

**Figure supplement 2—source data 2.** Table related to *Figure 2—figure supplement 2i*.

*supplement 1*). And, in vitro methylation assays showed that METTL3a, METTL3b, METTL3-Δ198, and METTL3-Δ238, similar to METTL3-WT, showed m⁶A methyltransferase activity (*Figure 2h–j*). These data suggest that these METTL3 variants have no effect on METTL3 localization or in vitro m⁶A methyltransferase activity.

It should be mentioned that the strong promoter CMV-driven pHAGE-METTL3-Flag construct used for purification of recombinant METTL3-Flag showed another highly expressed short form, named as METTL3c, which was very close in size to b (*Figure 2—figure supplement 2b*). Therefore, we had to exclude METTL3c expression for purification of METTL3b. We thus performed domain mapping for METTL3 with various mutations as indicated to identify conserved residues responsible for the expression of METTL3c, and finally found that point mutations in the M156 and M157 residues to convert these highly conserved amino acids into alanines (2MA) could lead to disappearance of METTL3c (*Figure 2—figure supplement 2c–g*). We then used this METTL3-2MA-Flag construct for edman sequencing as described in *Figure 2b*. METTL3c, which is disappear upon depletion of both M156 and M157, could be a product of an alternative open reading frame rather than a cleavage product, as deletion of the start codon leads to its loss. Since METTL3c was largely absent within endogenous METTL3, combined with data from m⁶A dot blots and tumor cell growth assays showing that METTL3-2MA functioned similarly to METTL3-WT (*Figure 2—figure supplement 2h–l*), we did not investigate METTL3c in the following study.

## Both METTL3a and FL METTL3 are required for cell proliferation and breast cancer progression

Since our above data have showed that METTL3a, METTL3b, METTL3-Δ198, and METTL3-Δ238 display the similar distributions with METTL3 WT and also harbor intact m⁶A methyltransferase activity in vitro, we wondered whether these two short forms, which are upregulated in breast cancer, are functionally important. We therefore reconstituted METTL3-WT, METTL3-Δ238, METTL3-Δ198, METTL3a, or METTL3b, respectively, in METTL3 knockout (KO) T47D and MDA-MB-231 cell lines (*Figure 3a, c*). Cell proliferation assays showed that reconstitution with METTL3-WT or METTL3-Δ198, but not METTL3-Δ238, METTL3a, or METTL3b, could rescue the defects in cell proliferation caused by METTL3 KO in T47D and MDA-MB-231 cell lines (*Figure 3b, d*). These findings suggest that the short form METTL3a and FL of METTL3 are required for tumor cell proliferation, since depletion of either one of them failed to rescue the cell growth defect caused by METTL3 KO. In line with these results, METTL3-Δ238 KI 293T cells displayed decrease in cell proliferation compared to METTL3 WT cells (*Figure 3e, f*), and reconstitution of METTL3a, but not METTL3b, could rescue the reduced cell proliferation (*Figure 3g, h*). These data suggest that METTL3a is required for METTL3-mediated regulation of cell proliferation.

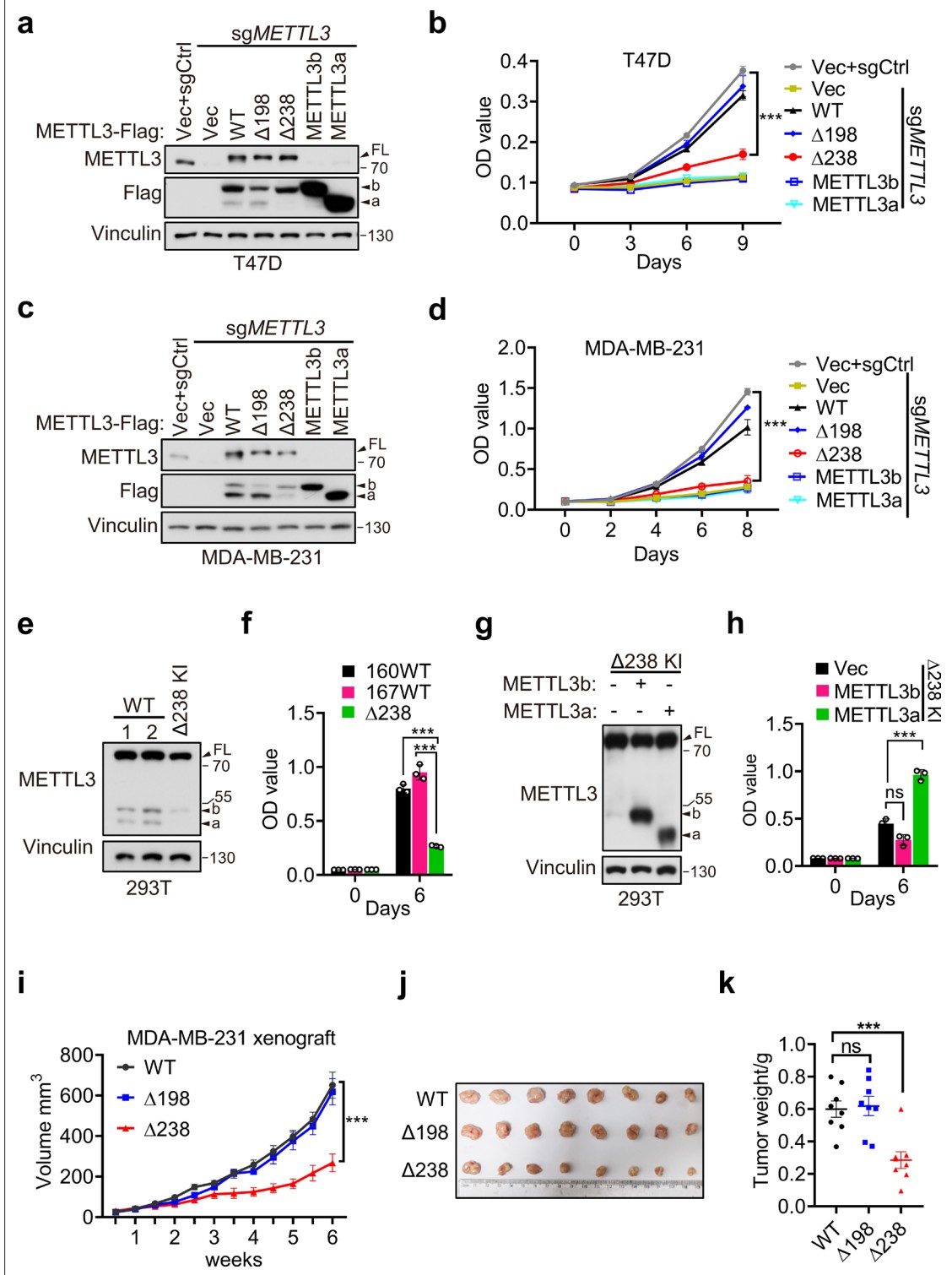

**Figure 3.** METTL3a contributes to cell proliferation and breast cancer progression. Immunoblot (**a**) and cell proliferation (**b**) of T47D cells infected with retrovirus encoding sg*METTL3* resistant METTL3 variants or control vector (Vec) followed by another infection with sg*METTL3* or sgControl (sgCtrl). Immunoblot (**c**) and cell proliferation (**d**) of MDA-MB-231 cells infected with retrovirus encoding sg*METTL3* resistant METTL3 variants or control vector (Vec) followed by another infection with sg*METTL3* or sgControl (sgCtrl). Immunoblot (**e**) and cell proliferation (**f**) of 293T cells with WT or CRISPR knock-in mediated deletion of Q238. Immunoblot analysis (**g**) and cell proliferation (**h**) of Δ238 knock-in 293T cells rescued with METTL3a or METTL3b. (**i–k**) Mouse xenograft experiments were performed with MDA-MB-231 cells infected with retrovirus encoding sg*METTL3* resistant METTL3-WT, METTL3-Δ198, or METTL3-Δ238 followed by another infection with sg*METTL3* (*n* = 8 mice per group). Tumor growth curve (**i**), tumor (**j**), and tumor

*Figure 3 continued on next page*

*Figure 3 continued*

weight (**k**) were recorded. Error bars represent mean ± standard error of the mean (SEM), unpaired *t*-test. ***p < 0.001, ns denotes no significance. FL indicates the full-length of METTL3. The short forms are labeled as a and b.

The online version of this article includes the following source data for figure 3:

**Source data 1.** Unedited western blot images for *Figure 3a, c, e, and g* and picture related to *Figure 3j*.

**Source data 2.** Tables related to *Figure 3b, d, f, h, i, and k*.

Further, we used METTL3-WT, METTL3-Δ238, or METTL3-Δ198 to, respectively, complement MDA-MB-231 METTL3 KO cells, and injected these cells orthotopically into the mammary fat pads of mice and performed bi-weekly measurement of tumor volume by caliper. Consistent with our in vitro results, we observed that reconstitution of either METTL3-WT or METTL-Δ198 could rescue the METTL3 KO-mediated defect in tumor growth, while expression of METTL3-Δ238 could not (*Figure 3i–k*). These results suggest that METTL3a is essential for METTL3-mediated regulation of breast cancer progression.

## METTL3a is required for the interaction between METTL3 and WTAP

METTL3-mediated tumor cell growth is known to rely on MTC for its methyltransferase activity (*Lan et al., 2019*). Here, our data showed that either METTL3 with loss of METTL3a (METTL3-Δ238) or METTL3a alone failed to rescue cell proliferation defects coursed by METTL3 KO, although both of them have intact catalytic activity in vitro. Therefore, we wondered whether METTL3a participates in the m⁶A MTC, among which METTL3, METTL14, and WTAP are core components required for intracellular RNA m⁶A modification (*Liu et al., 2014*; *Ping et al., 2014*). To this end, we performed immunoprecipitation (IP) assays in T47D and 293T cell lines and observed that METTL3-Δ238 had much weaker interaction with WTAP, but retained its interaction with METTL14, whereas

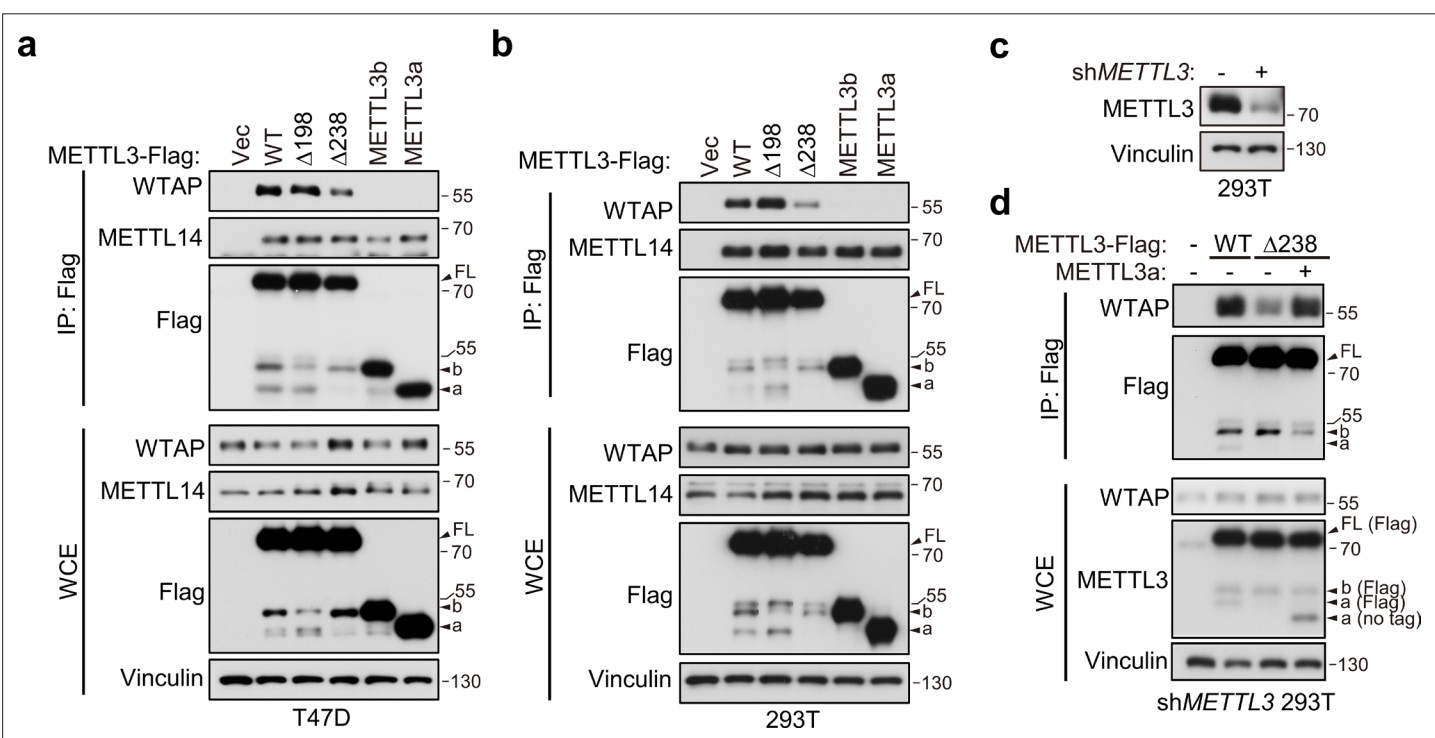

**Figure 4.** METTL3a is required for METTL3–WTAP interaction. Co-immunoprecipitation analyses of T47D cells (**a**) and 293T cells (**b**) transfected with Vector (Vec), METTL3-WT or its mutants. (**c**) Immunoblot analysis of 293T cells infected with or without lentivirus encoding *METTL3* shRNA. (**d**) Immunoblot of whole cell extracts (WCE) and immunoprecipitations (IP) of 293T cells infected with lentivirus encoding *METTL3* shRNA followed by transfection with METTL3-WT-Flag, METTL3-Δ238-Flag combining with or without METTL3a (no flag tag), or control vector as indicated.

The online version of this article includes the following source data for figure 4:

**Source data 1.** Unedited western blot images for *Figure 4*.

METTL3-Δ198 could interact with both WTAP and METTL14 (*Figure 4a, b*), consistent with the phenotypic study showing that METTL3-Δ238, but not METTL3-Δ198, affected tumor cell proliferation. Previous reports have shown that WTAP interacts with the METTL3 N-terminal leader helix domain (LH, 1–34aa), and that METTL14 can heterodimerize with METTL3 through their individual C-terminal methyltransferase domains (*Schöller et al., 2018*; *Wang et al., 2016a*; *Wang et al., 2016b*). In line with these findings, our results showed that both METTL3a and METTL3b, which lack the METTL3 N-terminal domain, failed to interact with WTAP, but could bind with METTL14 (*Figure 4a, b*). IP assays in METTL3 knockdown (KD) 293T cells further confirmed the decreased interaction between METTL3-Δ238 and WTAP, and reconstitution with exogenous METTL3a could rescue their interaction (*Figure 4c, d*), indicating that METTL3a is required for the METTL3–WTAP interaction.

## METTL3a mediates the METTL3–METTL3 interaction, a prerequisite step for WTAP recruitment in MTC complex

Since METTL3a showed no interaction with WTAP, we next sought to determine how reduced expression of METTL3a decreased the METTL3–WTAP interaction. Recent study has shown that METTL3 is capable of self-interaction and undergoes phase separation in nuclei (*Han et al., 2022*). Consistently, we also found that the METTL3–METTL3 interaction could occur in both 293T cell line and in vitro recombinant protein purification–IP systems (*Figure 5a, b*). And, the METTL3–METTL3 interaction is independent of DNA or RNA (*Figure 5c*). We further constructed METTL3 N-terminal (1–238) and C-terminal (239–580) truncation plasmids. And, IP assays showed that the METTL3–METTL3 interaction was through its C-terminal domain (*Figure 5d, e* and *Figure 5—figure supplement 1a–c*; *Su et al., 2022*; *Yan et al., 2022*). We therefore wondered whether METTL3a could mediate the METTL3–METTL3 interaction, then affects further recruitment of WTAP. Indeed, we found that the interaction between METTL3-Δ238 and METTL3-Δ238 was barely detectable, but restored under exogenous expression of METTL3a, as was its interaction with WTAP (*Figure 5f*), suggesting that METTL3a is required for the METTL3–METTL3 interaction, that is a prerequisite step for the METTL3–WTAP interaction. Our finding is in line with previous study showing that WTAP only interacts with METTL3 in dense phase, which is formed by METTL3–METTL3 interaction (*Han et al., 2022*). While METTL3b has no effect on the METTL3–METTL3 interaction (*Figure 5—figure supplement 1d*), which is consistent with our above data showing that METTL3b is dispensable for the METTL3–WTAP interaction.

We next examined the intracellular assembly process of MTC. Since METTL3 is the key which functions as methyltransferase, we first determined the effect of METTL3 on the formation of MTC. Through Co-IP assays with 293T cells, we found that METTL3 depletion decreased the METTL14–METTL14 and METTL14–WTAP interactions (*Figure 5g*), suggesting that the intracellular METTL14–METTL14 and METTL14–WTAP interactions take place at least partially through METTL3. While METTL14 depletion did not reduce, but rather increase the METTL3–METTL3 and METTL3–WTAP interactions (*Figure 5h*). Recent cryo-EM data demonstrate that WTAP can form dimer (*Su et al., 2022*). As expected, we observed the WTAP–WTAP interaction in 293T cells, and it was unaffected by depletion of METTL3 (*Figure 5i*). Conversely, WTAP depletion had no effect on the METTL3–METTL3 and METTL3–METTL14 interactions (*Figure 5j, k*). Altogether, we conclude that the METTL3–METTL3 interaction acts as a core initiating the formation of METTL3–METTL14–WTAP complex. These data demonstrate that METTL3a mediates the METTL3–METTL3 interactions, thereby forming larger complexes for efficient recruitment of WTAP in MTC.

WTAP is essential for m⁶A deposition (*Liu et al., 2014*; *Ping et al., 2014*), our current data suggest that METTL3a is required for METTL3–WTAP interaction. We therefore hypothesized that METTL3a may play a role in intracellular RNA m⁶A deposition. Through m⁶A dot blot assays, we observed that reconstitution of METTL3-Δ238 could not rescue the decreased RNA m⁶A modification caused by METTL3 depletion in 293T cells, while reconstitution of METTL3-Δ198 or METTL3-WT in these cells restored the m⁶A deposition (*Figure 5l, m*). Consistently, METTL3-Δ238 KI cells showed reduced m⁶A level compared to METTL3-WT cells, and reconstitution of exogenous METTL3a in METTL3-Δ238 KI cells rescued this decrease (*Figure 5n, o*). Collectively, our data suggest that METTL3a is required for the formation of METTL3–METTL14–WTAP complex and m⁶A RNA modification.

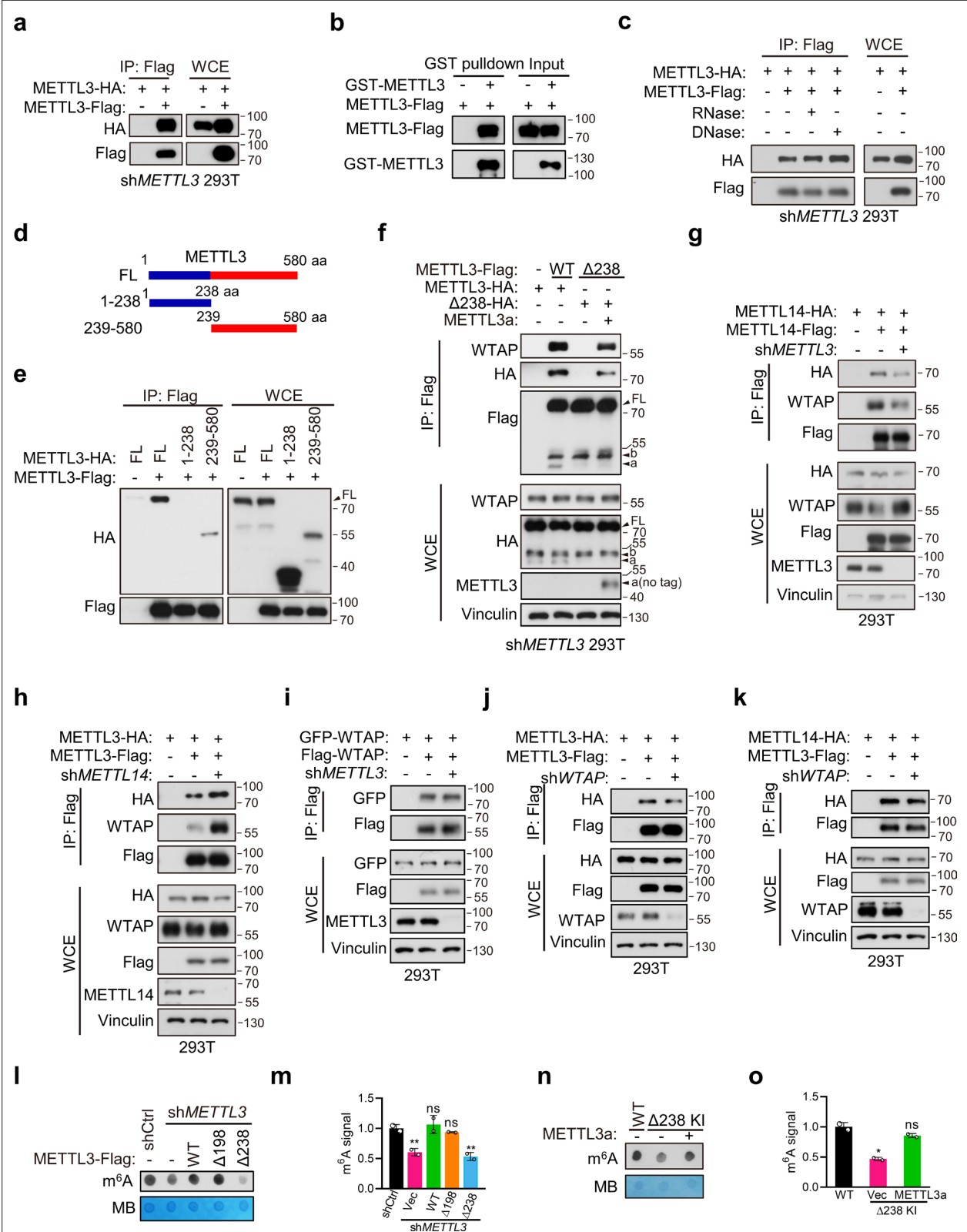

**Figure 5.** METTL3a mediates the METTL3–METTL3 interaction, a prerequisite step for WTAP recruitment in methyltransferase complex (MTC). (**a**) Immunoblot of whole cell extracts (WCE) and immunoprecipitations (IP) of 293T transfected with METTL3 containing different tag. (**b**) Immunoblot analysis showing the binding between purified METTL3-Flag and GST-METTL3. (**c**) Immunoblot of WCE and IP of 293T transfected with METTL3 containing different tag with or without RNase or DNase. (**d**) A schematic representation of METTL3 and its truncations used for IP. (**e**) Immunoblot of

*Figure 5 continued on next page*

*Figure 5 continued*

WCE and IP of 293T cells infected with lentivirus encoding *METLL3* shRNA followed by transfection with full-length (FL) or truncation of METTL3. (**f**) Immunoblot of WCE and IP of 293T cells infected with lentivirus encoding *METLL3* shRNA followed by transfection with METTL3-WT-flag, METTL3-Δ238-Flag combining with or without METTL3a (no flag tag) or control vector as indicated. Immunoblot of WCE and IP of 293T cells infected with or without lentivirus encoding *METLL3* shRNA (**g**) or *METTL14* shRNA (**h**) followed by transfection with METTL3 or METTL14 as indicated. (**i**) Immunoblot of WCE and IP of 293T cells infected with lentivirus encoding *METLL3* shRNA followed by transfection with WTAP containing different tag. (**j, k**) Immunoblot of WCE and IP of 293T cells infected with lentivirus encoding *WTAP* shRNA followed by transfection with METTL3 or METTL14 as indicated. $N^6$-methyladenosine ($m^6A$) dot blot (**l**) and quantification (**m**) of 293T cells infected with lentivirus encoding *METLL3* shRNA followed by transfection with METTL3-WT, METTL3 variants, or control vector as indicated. Methylene blue (MB) is used as a loading control. $m^6A$ dot blot (**n**) and quantification (**o**) of 293T cells with WT or CRISPR knock-in mediated deletion of Q238 with or without transfection of METTL3a. FL indicates the full-length of METTL3. The short forms are labeled as a and b. no tag represents exogenous METTL3a containing no flag tag.

The online version of this article includes the following source data and figure supplement(s) for figure 5:

**Source data 1.** Unedited western blot images for *Figure 5a–c, e–l, and n*.

**Source data 2.** Tables related to *Figure 5m and o*.

**Figure supplement 1.** The METTL3–METTL3 interaction depends on its C-terminal region.

**Figure supplement 1—source data 1.** Unedited western blot images for *Figure 5—figure supplement 1*.

## METTL3a is essential for the METTL3-mediated $m^6A$ deposition and mTOR activation

In light of our above data showing that METTL3a is required for $m^6A$ RNA methylation, we further examined how METTL3a affects $m^6A$ RNA modification across the whole genome in T47D cells. To this end, we expressed METTL3-WT, METTL3-Δ198, or METTL3-Δ238 to complement endogenous METTL3 KO T47D cells, then conducted $m^6A$-seq to identify their differentially modified transcripts. Mapping of transcripts with $m^6A$ modification revealed that $m^6A$ methylation was globally decreased from the transcriptome of METTL3 KO cells expressing empty vector (Vec) or METTL3-Δ238, while the total number of methylation marks was similar among METTL3-Δ198-, METTL3-WT-expressing METTL3 KO cells and the control (sgCtrl+Vec) cells (*Figure 6a*). Whereas motif analyses of the $m^6A$ peaks showed that the most enriched motif of the transcripts across all groups was the canonical GGAC motif (*Figure 6—figure supplement 1a*), and also the distribution of $m^6A$ methylation sites across the full transcriptome were similar among all groups (*Figure 6—figure supplement 1b*). To identify their differentially regulated $m^6A$ methylation sites, we first normalized their $m^6A$ peaks with that of METTL3 KO cells expressing vector (sg*METTL3*+Vec) to eliminate those peaks not affected by METTL3 KO, we then found that METTL3 KO cells expressing either METTL3-WT or METTL3-Δ198 shared the most of their $m^6A$ peaks (>80%), while those cells expressing METTL3-Δ238 lost 90% of peaks observed in METTL3-WT (*Figure 6b*). These analyses suggest that downregulation of METTL3a inhibits the METTL3-mediated $m^6A$ deposition.

To explore the potential targets regulated by METTL3a, we analyzed mRNAs associated with differentially regulated $m^6A$ peaks in METTL3-WT- versus METTL3-Δ238-complemented METTL3 KO cells and found that the differentially modified gene transcripts were enriched in several signal transduction pathways, and most highly in the mTOR pathway (*Figure 6c*). Previous studies have found that METTL3 promotes mTOR activity (*Chen et al., 2021*; *Qin et al., 2021*; *Sun et al., 2020a*), although the underlying mechanism has remained unclear.

In order to verify how METTL3a affects mTOR pathway, we analyzed the $m^6A$ methylome and found that some regulators in the mTOR pathway were methylated in METTL3-WT, but not in METTL3-Δ238 (*Figure 6d*). Immunoblot analysis of those well-established regulators for mTOR activation showed that TMEM127, a suppressor of mTOR (*Qin et al., 2010*), was significantly upregulated in METTL3 KO T47D cells (*Figure 6e*). From $m^6A$ sequencing data, we observed that METTL3 KO led to a significant decrease in $m^6A$ peaks at the 3′UTR of *TMEM127* transcript compared to that in cells reconstituted with METTL3-WT or METTL3-Δ198, but not METTL3-Δ238 (*Figure 6f*), which was further verified by $m^6A$ RIP assays in T47D and MDA-MB-231 cell lines (*Figure 6g* and *Figure 6—figure supplement 1c*). Subsequent immunoblot assays showed that TMEM127 was negatively regulated by METTL3, and a similar trend with the $m^6A$-seq data was also observed that METTL3-WT, but not METTL3-Δ238 expression could rescue TMEM127 expression in METTL3 KO T47D and MDA-MB-231 cell lines (*Figure 6h* and *Figure 6—figure supplement 1d*). Accordingly, METTL3 KO-mediated

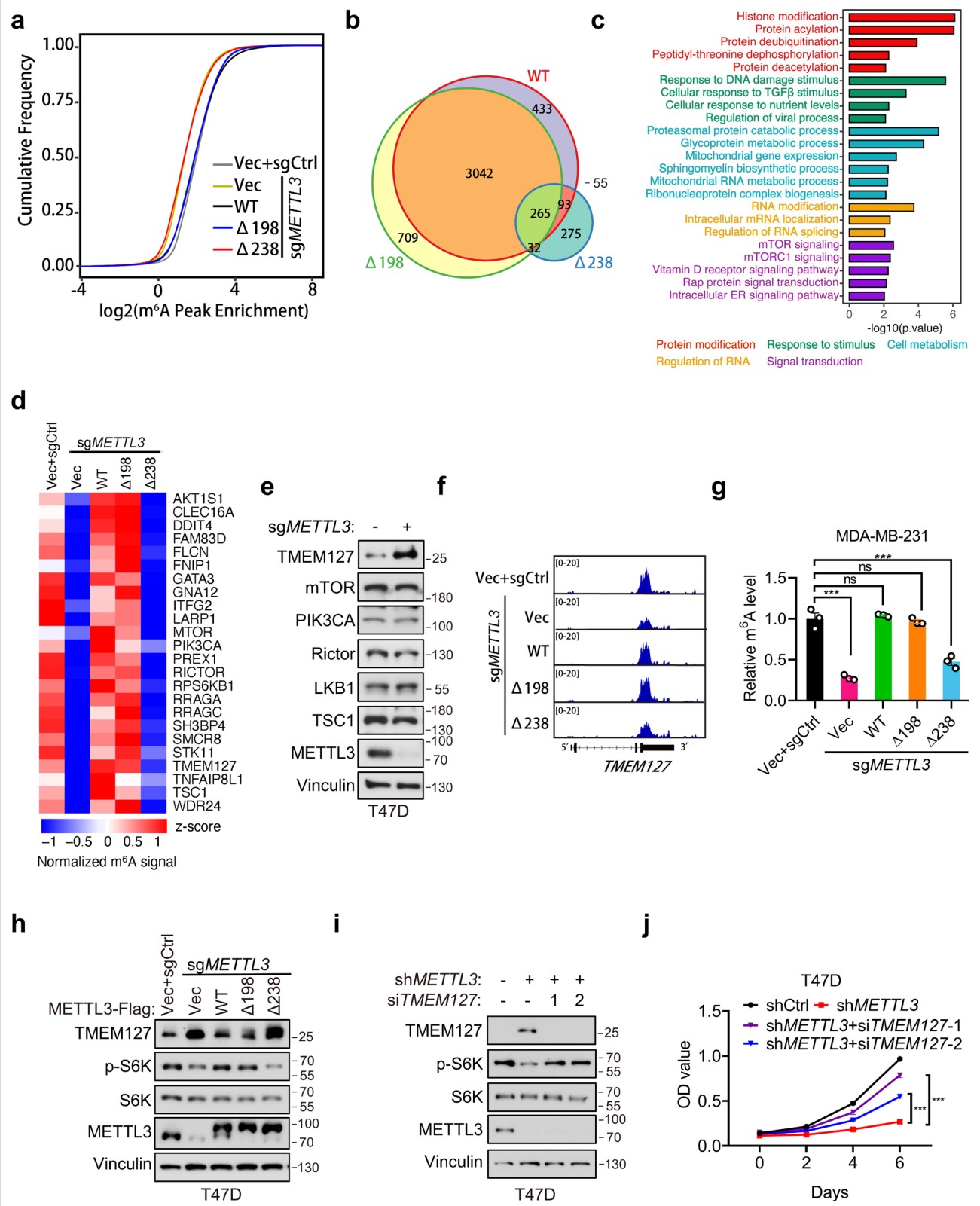

**Figure 6.** METTL3a is essential for the global METTL3-mediated *N*[6]-methyladenosine (m[6]A) deposition and mTOR activation. (**a**) Cumulative distribution function of log2 peak intensity of m[6]A-modified sites in T47D cells infected with retrovirus encoding sg*METTL3* resistant METTL3-WT, METTL3-Δ198, METTL3-Δ238, or control vector (Vec) followed by another infection with sg*METTL3* or sgControl (sgCtrl). (**b**) Overlaps of m[6]A peaks among METTL3-WT, METTL3-Δ198, and METTL3-Δ238 in METTL3 KO T47D cells. (**c**) GSEA with GO terms of differentially m[6]A methylated peaks in METTL3-WT and

*Figure 6 continued on next page*

*Figure 6 continued*

METTL3-Δ238 for molecular functions. (**d**) Heat map of m⁶A peaks related to mTOR pathway in T47D cells infected with retrovirus encoding sg*METTL3* resistant METTL3-WT, METTL3-Δ198, METTL3-Δ238, or control vector (Vec) followed by another infection with sg*METTL3* or sgControl (sgCtrl). (**e**) Immunoblot analysis of T47D cells infected with sgControl or sg*METTL3*. (**f**) Integrative genomic viewer (IGV) plots of m⁶A peaks at *TMEM127* mRNA in T47D cells as described in (**a**). (**g**) m⁶A-qPCR showing the m⁶A-enriched *TMEM127* transcripts from T47D cells. (**h**) Immunoblot of cell lysates from T47D cells as described in (**a**). Immunoblot (**i**) and cell proliferation (**j**) of T47D cells infected with sh*METTL3* or shControl followed by transfection with si*TMEM127* or siControl. Error bars represent mean ± standard error of the mean (SEM), unpaired *t*-test. \*\*\*p < 0.001, ns denotes no significance.

The online version of this article includes the following source data and figure supplement(s) for figure 6:

**Source data 1.** Unedited western blot images for *Figure 6e, h, and i*.

**Source data 2.** Tables related to *Figure 6c, d, g, j*.

**Figure supplement 1.** METTL3a is essential for the global METTL3-mediated *N*⁶-methyladenosine (m⁶A) deposition and mTOR activation.

**Figure supplement 1—source data 1.** Unedited western blot images for *Figure 6—figure supplement 1d*.

**Figure supplement 1—source data 2.** Table related *Figure 6—figure supplement 1c*.

decrease in S6K phosphorylation could be rescued by METTL3 WT or METTL3-Δ198, but not by METTL3-Δ238 (*Figure 6h* and *Figure 6—figure supplement 1d*). And, knockdown of TMEM127 in the METTL3 depleted T47D cells could restore S6K phosphorylation and cell proliferation (*Figure 6i, j*). Collectively, our findings demonstrate that METTL3a is required for mTOR activation at least partially through m⁶A-mediated suppression of TMEM127 expression, and this modulation contributes to tumor cell proliferation.

## Proteasome mediates METTL3 cleavage in an mTOR-dependent manner

In light of our above data showing that the short forms of METTL3 originated through post-translational regulation, we next investigated the mechanism and conditions responsible for processing FL METTL3 into short forms. It is well known that proteolytic cleavage mediates post-translational process of protein into small pieces, to identify the potential protease responsible for this process, we screened a series of inhibitors targeting different proteases and found that the proteasomal inhibitors MG132 and Ixazomib could largely block the generation of these short forms (*Figure 7a* and *Figure 7— figure supplement 1a*). Proteasome is multi-catalytic protease complex that regulates cellular protein homeostasis through targeted degradation of proteins (*Bochtler et al., 1999*), and previous work has described the endoproteolytic activity underlying proteasomal function (*Baugh and Pilipenko, 2004*; *Liu et al., 2003*; *Sorokin et al., 2005*). Here, we validated that MG132 treatment could reduce the levels of both endogenous and exogenous short forms of METTL3 in T47D and MDA-MB-231 cell lines (*Figure 7b–d*). Further, MG132 could also inhibit the in vitro proteolytic cleavage of purified recombinant METTL3-His incubated with lysates of T47D cells (*Figure 7e*). These results suggest that proteasome participates in METTL3 cleavage into the short forms.

Co-immunoprecipitation assays in 293T cells followed by liquid chromatography–tandem mass spectrometry (LC–MS)/MS showed that METTL3 interacted with the proteasome components PSMC3, PSMC5, and PSMD10 (*Figure 7f*), which was further confirmed by IP assays targeting METTL3-Flag in T47D cells (*Figure 7g*). In addition, knockdown of PSMC3, PSMC5, or PSMD10 abrogated the cleavage of METTL3 in MDA-MB-231 cells (*Figure 7h–j*). As PSMC3, PSMC5, and PSMD10 are all regulatory subunits of 26S proteasome, we hypothesized that 26S proteasome may be responsible for METTL3 cleavage. Indeed, in vitro proteasomal cleavage assay by incubation of purified recombinant METTL3 with 26S proteasome showed that 26S proteasome could directly cleave METTL3 WT to METTL3a and METTL3b, while deletion of both L198 and Q238 inhibited this process (*Figure 7k*). These results support the hypothesis that proteasome is responsible for the cleavage of METTL3 into METTL3a and METTL3b.

mTORC1 promotes the expression of proteasomal subunits (*Zhang et al., 2014*). We therefore investigated whether mTOR regulates the above proteasome components, and affects METTL3 cleavage. Western blot analysis showed that silence of mTOR in MDA-MB-231 cells inhibited the expression of PSMC3, PSMC5, and PSMD10 (*Figure 7l*), and also led to decreased accumulation of the METTL3 short forms (*Figure 7m*). Consistent with this finding, treatment with the mTOR inhibitor rapamycin in T47D and MDA-MB-231 cell lines led to decreased expression of these short forms

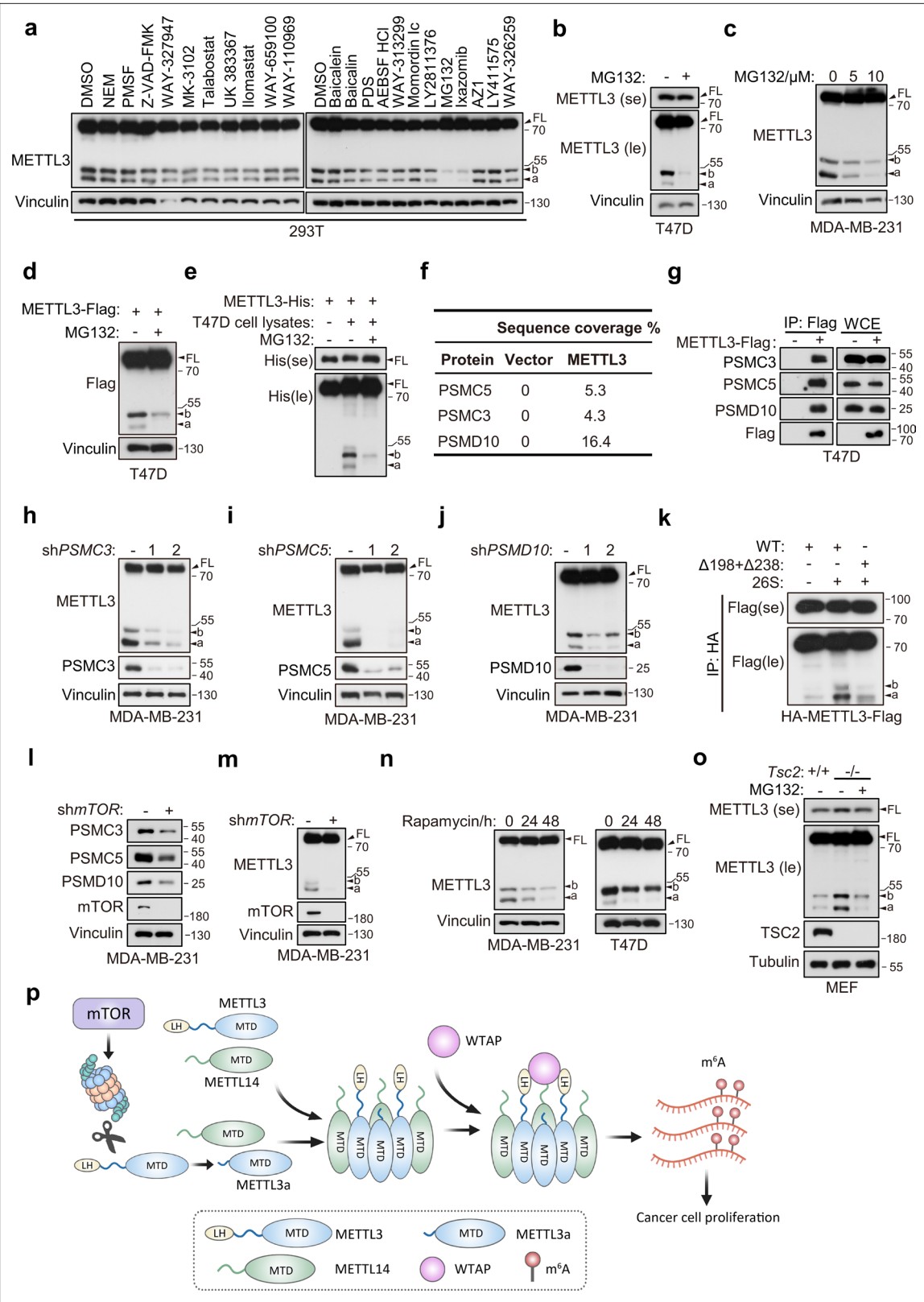

**Figure 7.** Proteasome mediates METTL3 cleavage in an mTOR-dependent manner. (**a**) Immunoblot of T47D cells treated with indicated inhibitors. Immunoblot of T47D (**b**) and MDA-MB-231 (**c**) cells treated with MG132 for 8 hr. (**d**) Immunoblot of T47D cells infected with METTL3-Flag followed by treatment with MG132 (10 μM) for 8 hr. (**e**) Immunoblot of purified recombinant METTL3-His protein incubated with or without T47D cell lysates wherein with or without MG132 treatment (20 μM) at 37°C for 1 hr. (**f**) List of the proteasome components identified by liquid chromatography–tandem

*Figure 7 continued on next page*

*Figure 7 continued*

mass spectrometry (LC–MS)/MS analysis of the tandem affinity purification with METTL3 antibody in 293T cells. (**g**) Immunoblot of whole cell extracts (WCE) and immunoprecipitations (IP) of T47D cells infected with METTL3-Flag or control vector. Immunoblot of MDA-MB-231 cells infected with shControl, sh*PSMC3* (**h**), sh*PSMC5* (**i**), and sh*PSMD10* (**j**). (**k**) Immunoblot of the purified HA-METTL3-Flag WT or indicated mutant from 293T cells using HA-magnetic beads followed by incubation with 26S at 37°C for 1 hr. (**l, m**) Immunoblot of MDA-MB-231 cells infected with sh*mTOR* or shControl. (**N**) Immunoblot of T47D cells or MDA-MB-231 cells with time course treatment of rapamycin (100 nM) as indicated. (**o**) Immunoblot of *Tsc2* WT (*Tsc2*[+/+]) or *Tsc2* KO (*Tsc2*[−/−]) MEF cells treated with or without MG132 (10 μM) for 8 hr. (**p**) A model for METTL3a-mediated methyltransferase complex (MTC) assembly. FL indicates the full-length of METTL3. The short forms are labeled as a and b. se and le indicated short exposure and long exposure, respectively.

The online version of this article includes the following source data and figure supplement(s) for figure 7:

**Source data 1.** Unedited western blot images for *Figure 7*.

**Figure supplement 1.** Proteasome mediates METTL3 cleavage in an mTOR-dependent manner.

**Figure supplement 1—source data 1.** Unedited western blot images for *Figure 7—figure supplement 1b and c*.

(*Figure 7n*). In addition, depletion of either Raptor or Rictor, which are key components of mTOR complexes 1 and 2 (mTORC1 and mTORC2), respectively, also reduced the expression of the METTL3 short forms (*Figure 7—figure supplement 1b, c*). We also found that cells with constitutively active mTOR signaling due to KO of *Tsc2*, a negative regulator of mTOR activity (*Crino et al., 2006*), led to upregulation of these short forms, and proteasome inhibitor MG132 treatment could inhibit this upregulation (*Figure 7o*), suggesting that mTOR upregulates the METTL3 short forms via proteasome. These cumulative results suggest that proteasome is responsible for METTL3 cleavage, and the mTOR pathway positively regulates this process.

## Discussion

In this study, we describe a cleaved form of METTL3, METTL3a, that is essential for the METTL3–WTAP interaction and m⁶A deposition, resulting in the promotion of breast tumor progression. Specifically, the cleaved form METTL3a mediates the METTL3–METTL3 interaction, which is a prerequisite step for WTAP binding with METTL3. Analyses of the RNA m⁶A methylome reveal that METTL3a is required for the global METTL3-mediated m⁶A modification and mTOR activation. Further, we find that the mTOR pathway positively regulates the expression of proteasome components that mediate proteasomal cleavage of METTL3 into METTL3a. These findings reveal a positive feedback between METTL3a and the mTOR signaling pathway (*Figure 7p*).

The MTC mediates m⁶A deposition. It is well established that the core components of the MTC include heterodimers of the METTL3–METTL14 methyltransferases with an adaptor protein WTAP. Here, we found that METTL3 undergoes METTL3–METTL3 interaction which is required for recruitment of WTAP in MTC. Downregulation of METTL3a leads to decrease in both METTL3–METTL3 interaction and the METTL3–WTAP interaction, and consequently attenuates m⁶A modification. Based on this work, we propose that METTL3a acts as a bridge between the METTL3–METTL14 heterodimers through mediating the METTL3–METTL3 interaction, thereby resulting in the formation of larger complexes which can efficiently recruit WTAP in MTC. Our finding is in line with previous report showing that METTL3 undergoes self-interaction (*Han et al., 2022*). In contrast to the constitutive binding of METTL14 to METTL3 in both the diffuse and the dense phase, WTAP only interacts with METTL3 in dense phase (*Han et al., 2022*), it means that WTAP tends to bind to larger METTL3–METTL14 complexes, which supports our conclusion. It is also noteworthy that METTL3b contains the region of residues 239–580 but is not required for the METTL3–WTAP interaction. Based on our findings, we are inclined to speculate that the cleavage product METTL3a is a structurally appropriate (i.e., small enough) component involved in MTC complex, whereas the METTL3b is not. Besides, overexpression of METTL3b in T47D, MDA-MB-231, or 293T cells did not show METTL3a expression in these cells (*Figure 3a, c, g*), indicating that METTL3b cannot be further cleaved to produce METTL3a, and the METTL3 cleavage may require its N-terminal region.

Screening of a proteasome inhibitor library in conjunction with LS–MS/MS, we found that the proteasome is responsible for METTL3 cleavage, and deletion of the proteasome components PSMC3, PSMC5, or PSMD10 attenuates the accumulation of cleaved forms of METTL3. Furthermore, we showed that the mTOR pathway promotes this process through positive regulation of proteasomal

components. The proteasome is essential for degradation of intracellular proteins, including misfolded mutant and damaged proteins (*Bochtler et al., 1999*). In addition, proteasome also exhibits endo-proteolytic activity (*Liu et al., 2003*), such as proteasome-mediated cleavage of translation initiation factors eIF4G and eIF3a differentially affects the assembly of ribosomal preinitiation complexes and thus inhibits the translation of viral RNA (*Baugh and Pilipenko, 2004*), and proteasomal cleavage of the Y-box-binding protein 1 splits off the C-terminal fragment which can translocate into nucleus and participate in cellular adaptation to DNA damage stress (*Sorokin et al., 2005*), which are similar to METTL3 cleavage identified in this work. Our data indicate that deletion of METTL3 residues L198 or Q238 does not completely block the formation of their respective short forms, leading us to speculate that the proteasome may recognize a conserved motif that likely contains multiple amino acids. It is also noteworthy that the cleavage ratios between METTL3a and METTL3b are different among those cell lines. For example, the ratio of METTL3a to METTLb was greater than 1 in MDA-MB-231 cells (*Figure 7c*), less than 1 in T47D and 293T cell lines (*Figure 7a, b*), and equal to 1 in MEF cells (*Figure 7o*), we speculate that there may be some factors that control the cleavage ratio between METTL3a and METTL3b, which warrants further investigation.

Other recent studies have demonstrated that mTOR increases global m⁶A modification by promoting WTAP translation (*Cho et al., 2021*), SAM synthesis (*Villa et al., 2021*), and by activating the chaperonin CCT complex (*Tang et al., 2021*). Here, we reveal a new mechanism by which mTOR promotes the cleavage of METTL3 through positive regulation of proteasome components, thereby enhancing global m⁶A deposition. METTL3-mediated tumorigenesis largely relies on its methyltransferase activity. The recent identification of a selective inhibitor of METTL3 catalytic activity that dramatically decreases tumor growth in mouse AML models suggests that m⁶A methylation by METTL3 could serve as a promising therapeutic target for cancer treatment (*Yankova et al., 2021*). Here, we show that the cleaved form METTL3a plays a key role in m⁶A methylation activity, thus providing another potential anti-cancer therapeutic approach through blockade of METTL3 cleavage. Several studies have demonstrated that proteasomal activity is upregulated in various cancers (*Arlt et al., 2009*; *Chen and Madura, 2005*), and several proteasome inhibitors have received regulatory approval and are routinely used in clinical settings (*Manasanch and Orlowski, 2017*). Our study expands the rationale for targeting the mTOR–proteasome axis in breast cancer therapies.

Previous studies have reported that METTL3 activates mTOR (*Chen et al., 2021*; *Qin et al., 2021*; *Sun et al., 2020a*), although the underlying mechanism has remained poorly defined. KEGG analysis of transcriptome-wide m⁶A sequencing data showed that the mTOR pathway is the most highly enriched signal transduction pathway affected by METTL3a depletion. Furthermore, we show that *TMEM127* is the primary target of METTL3 among those genes involved in the mTOR pathway, and depletion of METTL3a decreases m⁶A modification of the *TMEM127* 3′UTR, leading to elevated TMEM127 expression. TMEM127 serves as a suppressor of the mTOR pathway by interfering with mTOR accessibility for its regulators (*Qin et al., 2010*; *Yao et al., 2010*). Thus, mechanistically, METTL3a functions as a positive regulatory feedback for activation of the mTOR pathway.

In summary, this study represents the first description to our knowledge of cleaved forms of METTL3, with METTL3a in particular serving as an essential component of the MTC complex, and elucidates the intracellular assembly process of the METTL3–METTL14–WTAP complex. These findings also uncover a positive regulatory feedback between METTL3a and the mTOR pathway which may serve as the basis for development of therapeutic strategies for breast cancer.

## Materials and methods
### Cell culture and reagents
MDA-MB-231, T47D, and 293T cells obtained from American Type Culture Collection (ATCC). MEF (mouse embryo fibroblasts) *Tsc2* wild-type and KO cells were kind gift from Sheng-Cai Lin. MDA-MB-231 and 293T cells were cultured in Dulbecco's modified Eagle medium (Sigma-Aldrich) containing 10% fetal bovine serum plus 1% penicillin–streptomycin. T47D cells were cultured in RPMI-1640 (Sigma-Aldrich) containing 10% fetal bovine serum with 1% penicillin–streptomycin. Following virus infection, cells were maintained in the presence of G418 (100 μg/ml) or puromycin (2 μg/ml) depending on the vector. All cells were maintained in an incubator at 37°C and 5% $CO_2$. All cells were tested

with negatived mycoplasma contamination. Library of enzyme inhibitors were obtained from Selleck. Rapamycin was purchased from Sigma-Aldrich.

## Western blot and antibodies

EBC (epithelial buffer cell) buffer (50 mM Tris pH 8.0, 120 mM NaCl, 0.5% NP40, 0.1 mM EDTA (ethylenediaminetetraacetic acid), and 10% glycerol) supplemented with complete protease inhibitor (Roche Applied Biosciences) was used to harvest whole cell lysates. Cell lysate concentrations were measured by Bradford assay (Thermo Fisher Scientific), and equal amounts of proteins were loaded onto an SDS–polyacrylamide gel, separated by electrophoresis and blotted onto a nitrocellulose membrane (Milipore). Rabbit METTL3 (86132), HIF-1α (14179), p-S6K (9234), and S6K (34475) antibodies were obtained from Cell Signaling Technology. Antibodies against N-terminal METTL3 (A8070) and PSMC5 (A1538) were from ABclonal. Mouse Vinculin (V9131) and mouse Flag (F3165) antibody were purchased from Sigma-Aldrich. Antibodies against METTL14 (26158-1-AP), WTAP (mouse, 60188-1-Ig; rabbit, 10200-1-AP), PSMC3 (24142-1-AP), PSMD10 (12342-2-AP), mTOR (66888-1-Ig), TMEM127 (23142-1-AP), Raptor (20984-1-AP), Rictor (27248-1-AP), TSC1 (29906-1-AP), TSC2 (24601-1-AP), LKB1 (10746-1-AP), PIK3CA (27921-1-AP), and rabbit Flag (20543-1-AP) were obtained from Proteintech. Peroxidase-conjugated goat anti-mouse (170-6516) and peroxidase-conjugated goat anti-rabbit (1706515) secondary antibodies were purchased from Bio-Rad.

## Plasmids

The cDNA FL of METTL3 was amplified by PCR and subcloned into lentiviral vector pHAGE or retroviral vector pMSCV containing a C-terminal 3×FLAG or HA tag. KOD-Plus Mutagenesis Kit (SMK-101, TOYOBO) was used to construct METTL3 mutants. All plasmids were sequenced to confirm validity.

## siRNAs, lentiviral shRNA, and sgRNA vectors

Non-targeting siRNA was obtained from Dharmacon (D0012100220) as previous (*Zhang et al., 2018*). The siRNA target sequence of TMEM127 is as previously described (*Qin et al., 2010*). Lentiviral short-hairpin RNA (shRNA) against METTL3, METTL14, WTAP, mTOR, Rictor, Raptor, PSMC3, PSMC5, and PSMD10 were obtained by cloning into the pLKO.1 vector. The target sequences for METTL3 (CGTC AGTATCTTGGGCAAGTT), METTL14 (AAGGATGAGTTAATAGCTAAA), and WTAP (AAGGTTCGATTG AGTGAAACA) are as previously described (*Xiang et al., 2017*). The target sequences (CCGCATTG TCTCTATCAAGTT) for mTOR, Raptor (GGCTAGTCTGTTTCGAA ATTT), and Rictor (ACTTGTGAAGAA TCGTATCTT) are as previous reported (*Sarbassov et al., 2005*). The target sequences for PSMC3, PSMC5, and PSMD10 were obtained from Broad Institute TRC shRNA library. The target sequence is as follows:

shPSMC3-1: CCAAGCCATGAAGGACAAGAT (TRCN0000020229)
shPSMC3-2: CCAGCCCAACACCCAAGTTAA (TRCN0000020231)
shPSMC5-1: CAAGGTTATCATGGCTACTAA (TRCN0000020260)
shPSMC5-2: CAAACAGATCAAGGAGATCAA (TRCN0000020262)
shPSMD10-1: GAGTGCCAGTGAATGATAAAG (TRCN0000374678)
shPSMD10-2: CCGATAAATCCCTGGCTACTA (TRCN0000058073)

The sequence for METTL3 KO (AGAGTCCAGCTGCTTCTTGT) was as previously described (*Xiang et al., 2017*).

## Virus production and infection

293T packaging cell lines were used for lentiviral amplification. Lentiviral infection was carried out as previously described (*Zhang et al., 2018*). Briefly, lentiviral vector was co-transfected with pSPAX2 and pMD2.G in 293T cells using 40 kDa linearpolyethylenimine (Polysciences, 24,765). Retrovirus generation was by using PLAT-A cells (Cell Biolab). Viruses were collected at 48 and 72 hr post-transfection. After passing through 0.45 μm filters, viruses were used to infect target cells in the presence of 8 μg/ml polybrene (Sigma-Aldrich). Subsequently, target cell lines underwent appropriate antibiotic selection.

## Immunoprecipitation

Cells were solubilized with EBC buffer supplemented with complete protease and phosphatase inhibitors (Rhoche Applied Bioscience). Lysates were clarified by centrifugation and then mixed with

primary antibodies or anti-Flag M2 beads (Sigma-Aldrich) overnight. Bound complexes were washed with NETN (nuclear and cytoplasmic extraction) buffer (20 mM Tris pH 8.0, 100 mM NaCl, 0.5% NP40, 1 mM EDTA) six times and were eluted by boiling in SDS loading buffer. Bound proteins were resolved in SDS–PAGE followed by western blot analysis.

## Immunostaining

T47D cells were seeded on glass coverslip. Forty-eight hours later, the cells were fixed with 4% formaldehyde (Sigma-Aldrich) in phosphate-buffered saline (PBS) for 15 min at 37°C. After being washed with PBS three times, the cells were incubated with PBS plus 0.1% Triton X-100 buffer for 10 min. Next, the cells were blocked with PBS plus 10% fetal bovine serum at room temperature (RT) for 1 hr. Then, cells were incubated with the primary antibody for 2 hr at RT and subsequently washed with PBS buffer three times. Then, the cells were incubated with PBS buffer containing the secondary antibody (Jackson immunoresearch, Cy3 affinipure goat anti-rabbit IgG (H+L)), alexa fluor 488 affinipure donkey anti-mouse IgG (H+L) for 1 hr in the dark at RT. Finally, the cells were washed three times with PBS, and the slides were analyzed using a Leica confocal microscopy (Leica microsystem, Germany).

## Protein expression and purification from 293T cells

FL human METTL3-WT or mutant was cloned into pHAGE vector containing C-terminal 3×FLAG. The plasmid was transfected into 293T cells using PEI (Polysciences, high potency linear, MW = 40,000) transfection reagent. Protein purification from 293T cells was previously described (*Huang et al., 2021*). Briefly, cells were harvested after 36 hr transfection and washed three times with cold PBS. Then cells were resuspended with lysis buffer (50 mM HEPES (4-hydroxyethyl piperazine ethanesulfonic acid) at pH 7.4, 300 mM NaCl, 1 mM EGTA (Ethylene Glycol Tetraacetic Acid), 0.5% Triton X-100 and complete protease inhibitor) and incubated on ice for 15 min. Then centrifuged at 14,000 rpm for 20 min at 4°C, and the supernatant was block with protein G agarose for 1 hr at 4°C. Next the supernatant was incubated the anti-Flag M2 affinity Gel (Sigma-Aldrich) for 4 hr at 4°C. The gel was washed with three times with lysis buffer and washed twice with wash buffer (50 mM HEPES at pH 7.4, 150 mM NaCl, 1 mM EGTA, 0.5% Triton X-100). Then the IP product was eluted with 3×FLAG peptide. Last the elution products were stored at −80°C or loaded onto an SDS–polyacrylamide gel and blotted onto PVDF (polyvinylidene fluoride) membrane (Milipore) and stained with *Ponceau S* for Edman sequencing.

## Liquid chromatography and mass spectrometry

The eluate from IP was resolubilized in 8 M urea, then each sample was reduced with 5 mM DTT (dithiothreitol) and alkylated (50 mM IAA (iodoacetic acid), 30 min, RT in the dark). After trypsin digestion overnight at 37°C, the peptide was desalted by using C18 StageTip columns and resolubilized in 0.1% formic acid. The LC–MS/MS is equipped with an Easy nLC 1200 (Thermo Scientific) coupled to the QExactive HF (Thermo Scientific). Samples were injected onto an Easy Spray PepMap C18 column (75 µm × 150 mm, 3 µm, C18, Dr. Maisch GmbH) and separated over a 120-min method. The gradient for separation consisted of 2–100% buffer B at a 300-nl/min flow rate, where buffer A was 0.1% formic acid in water and buffer B consisted of 0.1% formic acid in ACN. After peptide separation, the QExactive HF was operated in data-dependent mode. Resolution for the precursor scan (*m/z* 200) was set to 60,000, while MS/MS scans resolution was set to 15,000. Raw data were processed using MaxQuant 1.6.1.0 and MS/MS spectra were correlated against the human Uniprot database (uniprot-*Homo sapiens* [9606]–192901–20201111.fasta). The mass spectrometry proteomics data have been deposited to the ProteomeXchange Consortium via the PRIDE (*Perez-Riverol et al., 2022*) partner repository with the dataset identifier PXD036899.

## In vitro METTL3 cleavage assay

The expressing of METTL3-His was according to the previous described (*Huang et al., 2019*). T47D cells were incubated with cold buffer (50 mM HEPES at pH 7.4, 150 mM NaCl, 1 mM DTT) for 15 min at 4°C. And then cells were disrupted by ultrasonication. Cell lysates were obtained by centrifuging twice. Incubations of 1 µg proteins with 10 µl cell lysates were performed at 37°C for 1 hr and reactions were stopped by addition of SDS loading buffer.

## Proteolytic assay

HA-METTL3-Flag protein was purified from 293T by using anti-HA magnetic beads (Sigma-Aldrich) as described in this method. Proteasomal cleavage assay was performed as described (*Liu et al., 2003*). Briefly, incubations of 1 µg proteins with 0.5 µg 26S proteasome (Lifersensors) were performed at 37°C for 1 hr in a total volume of 20 µl of buffer B (20 mM Tris–HCl, pH 7.1, 20 mM NaCl, 10 mM MgCl$_2$, 0.25 mM ATP, and 1 mM DTT). Reactions were stopped by addition of SDS loading buffer.

## GST protein purification and GST pull-down

Glutathione *S*-transferase plasmids were transformed with BL21 competent cells. Single colonies were picked from above and cultured in 50 ml LB medium containing ampicillin. After overnight culture, 5 ml LB medium was diluted in 500 ml LB medium for shaking at 37°C for 2–3 hr until OD600 of 0.8–1.0 was reached. 0.2 mM IPTG (isopropyl-β-d-thiogalactoside) was added to induce GST protein production at 4°C for 16 hr before harvesting pellets. Bacteria lysates were disrupted by the nanodebee homogenizer. Cleared bacteria lysates were purified by using glutathione–Sepharose 4B beads. About 10 µl of GST suspension proteins was incubated with purified protein in 500 µl NETN buffer. After 4 hr incubation, bound complexes were washed with NETN buffer eight times followed by boiling in SDS loading buffer and SDS–PAGE.

## Cell proliferation assay

Cells were plated in triplicate in 96-well plates (2000 cells/well) in appropriate growth medium. At indicated time points, cells were replaced with 90 µl fresh growth medium supplemented with 10 µl MTT (methyl thiazolyl tetrazolium) reagents (Promega), followed by incubation at 37°C for 2 hr. OD absorbance values were measured at 490 nm using a 96-well plate reader (Biotech).

## Orthotopic tumor growth

Six-week-old female nude mice were used for xenograft studies. Approximately 1 × 10$^6$ viable MDA-MB-231 breast cancer cells were resuspended in 1:1 ratio in 50 µl medium and 50 µl matrigel (Corning, 354234) and injected orthotopically into the fourth mammary fat pad of each mouse. After injection, tumor size was measured twice a week using an electronic caliper. Tumor volumes were calculated with the formula: volume = $(L \times W^2)/2$, where $L$ is the tumor length and $W$ is the tumor width measured in millimeters. The total mass of tumors was presented as mean ± standard error of the mean (SEM) and evaluated statistically using the unpaired two-tailed Student's *t*-test. All mouse experiments were approved by the Animal Care Committees of Wuhan University Medical Research Institute (Wuhan University Center for Animal Experiment/ABSL-III Laboratory; approval number WQ20210109).

## Human tumors

Fresh-frozen samples of breast tumor and their accordingly adjacent normal tissues were obtained from Zhongnan Hospital of Wuhan University. All experiments involving human samples were conducted in compliance with all relevant ethical regulations and were approved by the Medical Ethics Committees of School of Medicine, Wuhan University (approved number: ChiCTR1800014247).

## m$^6$A dot blot

Total RNA was extracted by using MiniBEST Universal RNA extraction Kit (TaKaRa) according to the manufacturer's instructions and quantified by NanoDrop instruments (Thermo Fisher). After denaturation, RNA was dropped onto the Amersham Hybond-N+ membrane (RPN119B, GE Healthcare) and UV crosslinked to the membrane. Then the membrane was blocked 1 hr with 5% nonfat milk, and incubated with specific anti-m$^6$A (1:1000 dilution, Synaptic Systems, 202003) antibody overnight at 4°C. Wash three times with TBST ( tris buffered saline tween) and incubated with HRP-conjugated ( horseradish peroxidase) goat anti-rabbit IgG (Bio-Rad).

## m$^6$A methyltransferase activity assay

The in vitro methyltransferase activity assay was performed following the published procedure (*Li et al., 2016*; *Yan et al., 2022*). The METTL3-WT or mutants and METTL14 were co-expressed in 293T cells lysed by using high salt buffer (50 mM HEPES at pH 7.4, 300 mM NaCl, 1 mM EGTA, 0.5% Triton X-100 and complete protease inhibitor) and co-purified by Anti-FLAG M2 Magnetic Beads (Sigma-Aldrich).

Briefly, a typical 50 µl of reaction mixture containing the following components: 1 µM RNA probes (probe 1: ACGAGUCCUGGACUGAAACGGACUUGU, probe 2: ACGAGUCCUGGAUUGAAACG GAUUUGU), 1 µM co-purified METTL3-WT-Flag and Flag-METTL14, or co-purified METTL3 mutants-Flag and Flag-METTL14, 1 µM SAM, 0.25 U/µl RNasin, 1 mM DTT, 0.01% Triton X-100, and 20 mM Tris–HCl (pH 7.5). The reactions were incubated at 23°C for 1 hr and were quenched by the addition of trifluoroacetic acid to a final concentration of 0.1% (vol/vol). The methylation activity was measured using dot blot with the m⁶A antibody (1:300 dilution, Synaptic Systems, 202003) or the Promega biolu-minescence assay (MTase-Glo) through detecting by-product SAH.

## m⁶A-seq

Total RNA was extracted using Trizol reagent (Invitrogen, CA, USA) following the manufacturer's procedure. The total RNA quality and quantity were analysis of Bioanalyzer 2100 and RNA 6000 Nano LabChip Kit (Agilent, CA, USA) with RIN (RNA integrity number) number >7.0. Following purification with poly-T oligo attached magnetic beads (Invitrogen), the poly(A) mRNA fractions are fragmented, and then were subjected to incubated for 2 hr at 4°C with m⁶A-specific antibody (No. 202003, Synaptic Systems, Germany). Eluted m⁶A-containing fragments (IP) and untreated input control fragments are converted to final cDNA library in accordance with a strand-specific library preparation by dUTP method. Then we performed the paired-end 2 × 150 bp sequencing on an Illumina Novaseq 6000 platform at the LC-BIO Bio-tech Ltd (Hangzhou, China) following the vendor's recommended protocol. The data have been deposited in the GEO repository with the accession numbers.

## Gene-specific m⁶A qPCR

m⁶A qPCR was performed as previous described (*Li et al., 2017b*). Briefly, total RNA was isolated with MiniBEST Universal RNA extraction Kit (TaKaRa). 500 ng total RNA was saved as input sample, the rest mRNA was used for m⁶A-immunoprecipitation. Nearly, 200 µg total RNA were incubated with m⁶A anti-body (No. 202003, Synaptic Systems, Germany) and diluted into 500 µl IP buffer (150 mM NaCl, 0.1% NP-40, 10 mM Tris–HCl, pH 7.4, 100 U RNase inhibitor), and then rotated at 4°C for 2 hr, following mixed with Dynabeads M-280 and rotated for another 2 hr. After 4 times washing by IP buffer, the IP portion were eluted with 300 µl Proteinase K buffer (5 mM Tris–HCl pH 7.5, 1 mM EDTA pH 8.0, 0.05% SDS, and 4.2 µl Proteinase K [20 mg/ml]) for 1.5 hr at 50°C (*Meyer et al., 2012*), and RNA was recovered with phenol:chloroform extraction followed by ethanol precipitation. The final RNA was reverse transcribed by using ReverTra Ace qPCR RT Kit (TOYOBO). The *TMEM127* mRNA level was determined by the number of amplification cycles (Cq). The relative m⁶A levels in *TMEM127* were calculated by the m⁶A levels (m⁶A IP) normalized using the expression of Input. The qPCR primer is as follows:

5'-AATTCAGCCAGACCCAGAGC-3', 5'-GCATCAGACCCACACTGTCA-3'

## Bioinformatics analyses

Raw sequencing files were filtered using cutadapt (v1.18), and then aligned against the human reference genome (GRCh38/hg38) using tophat (v2.1.1) (*Kim et al., 2013*). The m⁶A-enriched regions in each m⁶A-immunoprecipitation sample were identified by MACS2 (v2.2.6) with the option of '--nomodel' with corresponding input library as control (*Zhang et al., 2008*). Peaks identified with p < 1e−10 were used for downstream analysis. Tracks of signal were computed using MACS2 bdgcmp module with parameter '-m FE', and then visualized using the Integrative Genomics Viewer (IGV) visualization tool (v2.4.14). Peak annotation and motif search were both accomplished using the soft-ware HOMER (v4.11) (*Heinz et al., 2010*). De novo motifs with 6 bp length were detected within 100 bp around m⁶A peak summits identified by MACS2. Differential m⁶A-modified peaks were iden-tified using exomePeak2 (v1.6.1) (p < 0.05). Metagene analysis of m⁶A distribution on transcripts was performed using the MetaPlotR pipeline. GO term enrichment analysis was performed by R packages clusterProfiler (v3.8.1) (*Wu et al., 2021*). The enriched GO terms were further subjected to the web tool REVIGO with the default parameter to filter redundant terms (*Supek et al., 2011*).

## Statistical analysis

Unpaired two-tailed Student's *t*-test was used for experiments comparing two sets of data. Data represent mean ± SEM from three independent experiments. *, **, and *** denote p value of <0.05, 0.01, and 0.001, respectively. ns denotes no significance.

## Acknowledgements

The authors thank members of our laboratory for helpful discussions. We thank Dr. Qing Zhang (University of Texas Southwestern), Dr. Pengda Liu (University of North Carolina at Chapel Hill), Dr. Kai Jiang (Wuhan University), Dr. Rui Xiao (Wuhan University), Dr. Yanxun Yu (Wuhan University), Dr. Youngnam Jin (Wuhan University), Dr. Ying Zhang (Wuhan University), and Dr. Xiaodong Zhang (Wuhan University) for critical readings and helpful suggestions. The authors thank Dr. Hong-Bing Shu (Wuhan University) for providing pMSCV emptor vector and METTL14 plasmids and Dr. Chen-Song Zhang (Xiamen University) for providing Tsc2−/− MEF cells. This work was supported by the Fundamental Research Funds for the Central Universities [2042020kf0197 and 2042022dx0003 to JZ]; National Natural Science Foundation of China [31970737 to JZ and 32100570 to CY]; the Startup Funding from Wuhan University [to JZ]; Natural Science Foundation of Hubei Province [2020CFA071 and 2022CFA008 to JZ]; the National Key Research and Development Program of China [2022YFA1305400 to JZ]; and the China Postdoctoral Science Foundation [2020M672408 to CY]. We also sincerely thank the core facility of the Medical Research Institute at Wuhan University for their technical support.

## Additional information

### Funding

| Funder | Grant reference number | Author |
|---|---|---|
| Fundamental Research Funds for the Central Universities | 2042020kf0197 | Jing Zhang |
| Fundamental Research Funds for the Central Universities | 2042022dx0003 | Jing Zhang |
| National Natural Science Foundation of China | 31970737 | Jing Zhang |
| National Natural Science Foundation of China | 32100570 | Chaojun Yan |
| The startup funding from Wuhan University | | Jing Zhang |
| Natural Science Foundation of Hubei Province | 2020CFA071 | Jing Zhang |
| National Key Research and Development Program of China | 2022YFA1305400 | Jing Zhang |
| China Postdoctoral Science Foundation | 2020M672408 | Chaojun Yan |
| Natural Science Foundation of Hubei Province | 2022CFA008 | Jing Zhang |

The funders had no role in study design, data collection, and interpretation, or the decision to submit the work for publication.

### Author contributions

Chaojun Yan, Jing Zhang, Funding acquisition; Jingjing Xiong, JX, Assisted with experiments and data analysis; Zirui Zhou, Z.R., Assisted with experiments and data analysis; Qifang Li, Q.L., Assisted

with experiments and data analysis; Chuan Gao, C.G., performed the bioinformatics analyses; Mengyao Zhang, Y.Z., helped with plasmid construction; Liya Yu, L.Y., helped with animal experiment; Jinpeng Li, JL, provided essential reagents; Ming-Ming Hu, M.H. provided the key advices on the project; Chen-Song Zhang, C.Z., provided the key advices on the project; Cheguo Cai, C.C., provided the key advices on the project; Haojian Zhang, H.Z., provided the key advices on the project

### Author ORCIDs
Jing Zhang https://orcid.org/0000-0001-6438-9113

### Ethics
All experiments involving human samples were conducted in compliance with all relevant ethical regulations and were approved by the Medical Ethics Committees of School of Medicine, Wuhan University (approved number: ChiCTR1800014247).
All mouse experiments were approved by the Animal Care Committees of Wuhan University Medical Research Institute (Wuhan University Center for Animal Experiment/ABSL-III Laboratory; approval number WQ20210109). All surgery was performed under isoflurane anesthesia, and every effort was made to minimize suffering.

Reviewer #1 (Public Review): https://doi.org/10.7554/eLife.87283.3.sa1
Reviewer #2 (Public Review): https://doi.org/10.7554/eLife.87283.3.sa2
Author Response https://doi.org/10.7554/eLife.87283.3.sa3

## Additional files

### Supplementary files
• MDAR checklist

### Data availability
The $m^6$A-seq data have been deposited in the GEO repository under the accession numbers GEO (GSE213727). The LC-–MS/MS data have been deposited in ProteomeXchange with identifier (PXD036899).

The following datasets were generated:

| Author(s) | Year | Dataset title | Dataset URL | Database and Identifier |
|---|---|---|---|---|
| Yan C, Xiong J, Zhou Z, Li Q, Gao C, Zhang M, Yu L, Li J, M-M Hu, Zhang C-S, Cai C, Zhang H, Zhang J | 2023 | A Cleavage Product METTL3a-(238-580) Mediates m6A Methyltransferase Complex Assembly and mTOR Activation | https://www.ncbi.nlm.nih.gov/geo/query/acc.cgi?acc=GSE213727 | NCBI Gene Expression Omnibus, GSE213727 |
| Yan C, Xiong J, Zhou Z, Li Q, Gao C, Zhang M, Yu L, Li J, M-M Hu, Zhang C-S, Cai C, Zhang H, Zhang J | 2023 | A cleaved METTL3 potentiates the METTL3–WTAP interaction and breast cancer progression | http://proteomecentral.proteomexchange.org/dataset/PXD036899 | ProteomeXchange, PXD036899 |

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
