## [Editor Report · eLife assessment]

This study presents the **valuable** finding that a cleaved form of METTL3 (termed METTL3a) has an essential role in regulating the assembly of the METTL3-METTL14-WTAP complex. The evidence supporting the claims of the authors is **solid**, and the work will be of interest to medical biologists working on breast cancer.

---

## [Referee Report · Reviewer #1 (Public Review)]

In mammals, a large methyltransferase complex (including METTL3, METTL14 and WTAP) deposits m6A across the transcriptome, and METTL3 serves as its catalytic core component. In this manuscript, the authors identified two cleaved forms of METTL3 and described the function of METTL3a (residues 239-580) in breast tumorigenesis. METTL3a mediates the assembly of METTL3-METTL14-WTAP complex, the global m6A deposition and breast cancer progression. Furthermore, the METTL3a-mTOR axis was uncovered to mediate the METTL3 cleavage, providing potential therapeutic target for breast cancer. This study is properly performed and the findings are very interesting; however, some problems with the model and assays need to be modified.. It is widely known that METTL3 and METTL14 form a stable heterodimer with the stoichiometric ratio of 1:1 (Wang X et al. Nature 534, 575-578 (2016), Su S et al. Cell Res 32(11), 982-994 (2022), Yan X et al. Cell Res 32(12), 1124-1127 (2022)), the numbers of METTL3 and METTL14 in the model of Fig 7P are not equivalent and need to be modified.

---

## [Referee Report · Reviewer #2 (Public Review)]

In this study, Yan et al. report that a cleaved form of METTL3 (termed METTL3a) plays an essential role in regulating the assembly of the METTL3-METTL14-WTAP complex. Depletion of METTL3a leads to reduced m6A level on TMEM127, an mTOR repressor, and subsequently decreased breast cancer cell proliferation. Mechanistically, METTL3a is generated via 26S proteasome in an mTOR-dependent manner.

The manuscript follows a smooth, logical flow from one result to the next, and most of the results are clearly presented. Specifically, the molecular interaction assays are well-designed. This model represents a significant addition to the current understanding of m6A-methyltransferase complex formation.

---

## [Author Response]

The following is the authors’ response to the original reviews.

**Reviewer #1 (Public Review):**
This study presents an important finding on human m6A methyltransferase complex (including METTL3, METTL14 and WTAP). The evidence supporting the claims of the authors is convincing, although the model and assays need to be further modified. The work will be of interest to biologists working on RNA epigenetics and cancer biology.In mammals, a large methyltransferase complex (including METTL3, METTL14 and WTAP) deposits m6A across the transcriptome, and METTL3 serves as its catalytic core component. In this manuscript, the authors identified two cleaved forms of METTL3 and described the function of METTL3a (residues 239-580) in breast tumorigenesis. METTL3a mediates the assembly of METTL3-METTL14-WTAP complex, the global m6A deposition and breast cancer progression. Furthermore, the METTL3a-mTOR axis was uncovered to mediate the METTL3 cleavage, providing potential therapeutic target for breast cancer. This study is properly performed and the findings are very interesting; however, some problems with the model and assays need to be modified. It is widely known that METTL3 and METTL14 form a stable heterodimer with the stoichiometric ratio of 1:1 (Wang X et al. Nature 534, 575-578 (2016), Su S et al. Cell Res 32(11), 982994 (2022), Yan X et al. Cell Res 32(12), 1124-1127 (2022)), the numbers of METTL3 and METTL14 in the model of Fig 7P are not equivalent and need to be modified.

We thank for reviewer’s good suggestion. We have modified the model in Fig. 7P.

**Reviewer #2 (Public Review):**
In this study, Yan et al. report that a cleaved form of METTL3 (termed METTL3a) plays an essential role in regulating the assembly of the METTL3-METTL14-WTAP complex. Depletion of METTL3a leads to reduced m6A level on TMEM127, an mTOR repressor, and subsequently decreased breast cancer cell proliferation. Mechanistically, METTL3a is generated via 26S proteasome in an mTOR-dependent manner.The manuscript follows a smooth, logical flow from one result to the next, and most of the results are clearly presented. Specifically, the molecular interaction assays are welldesigned. If true, this model represents a significant addition to the current understanding of m6A-methyltransferase complex formation.A few minor issues detailed below should be addressed to make the paper even more robust. The specific comments are contained below.1. The existence of METTL3a and METTL3b.

We thank reviewer for point this out. We discovered the cleaved form of METTL3 in breast cancer, and we further examined this cleaved METTL3 in other cell lines such as lung cancer cell lines, renal cancer cell lines, HCT116 and MEF (new Supplementary Figures 1A-1C), these data suggest that it is a common rule. Therefore, we speculate that METTL3a may be ubiquitiously expressed. We have added this part in the revised manuscript, please see Line 118-120.

1. Generation of METTL3a and METTL3b.1. Figure 1 shows that METTL3a and METTL3b were generated from the C-terminal of full-length METTL3. Because the sequence of METTL3a is involved in the sequences of METTL3b, can METTL3b be further cleaved to produce METTL3a?

Although the sequence of METTL3a is involved in the sequences of METTL3b, overexpression of METTL3b in T47D, MDA-MB-231 and 293T cells did not show METTL3a expression in these cells (please see Figures 3A, 3C, 3G), suggesting that METTL3b can not be further cleaved to produce METTL3a, and the METTL3 cleavage may require its N-terminal region. We have added this in the discussion, please see Line 358 to 360.

1. Based on current data, the generation of METTL3a and METTL3b are separated. Are there any factors that affect the cleavage ratio between METTL3a and METTL3b?

We thank for reviewer’s excellent question. In this study, we show that both METTL3a and METTLb are produced through proteasomal cleavage, and both of them are positively regulated by the mTOR pathway. On the other hand, we indeed observed the differential cleavage ratios between METTL3a and METTL3b across different cell lines. For example, METTL3a/METTLb ratio was greater than 1 in MDA-MB-231 cells (see Figure 7C), less than 1 in T47D and 293T cell lines (see Figure 7A and 7B), and equal to 1 in MEF cells (see Figure 7O). Based on these results, we speculate that there may be some factors that control the cleavage ratio between METTL3a and METTL3b, which warrants further investigation. We have added this in the discussion, please see Line 374 to 379.

1. In Figure 2G, the author shows the result that incubation of the Δ198+Δ238 METTL3 protein with T47D cell lysates cannot produce the METTL3a and METTL3b variants. The author may also show the results that Δ198 METTL3 protein or Δ238 METTL3 protein incubates with T47D cell lysates, respectively.

Following the reviewer’s suggestion, we had performed in vitro cleavage assays by incubation of METTL3-Δ238 or METTL3-Δ198 with T47D cell lysates, and had incorporated this result in the revised manuscript. Please see our new Supplementary Figure 3A.

1. As well as many results published in previous studies, the in vitro methylation assay shows that WT METTL3 is capable of methylating RNA probe (figure 2H). The main point of this study is that METTL3a is required for the METTL3-METTL14 assembly. However, the absence of METTL3a in the in vitro system did not inhibit METTL3METTL14 methylation activity. Moreover, the presence of METTL3a even resulted in a weak m6A level.

The main point of this study is that METTL3a is required for the METTL3WTAP interaction, but dispensable for the METTL3-METTL14 assembly (see Figure 4A-4B). In this in vitro methylation assays, METTL3 and METTL14 is capable of methylating RNA probe in the absent of WTAP. In this condition, we found that METTL3 WT as well as its different variants (METTL3-Δ238, METTL3-Δ198, METTL3b and METTL3a) except the catalytically dead mutant METTL3 APPA showed methylation activity in vitro.

1. In Figure 4A, the author suggests that WTAP cannot be immunoprecipitated with METTL3a and 3b because WTAP interacted with the N-terminal of METTL3. If this assay is performed in WT cells, the endogenous full-length METTL3 may help to form the complex. In this case, WTAP is supposed to be co-immunoprecipitated.

We thank reviewer for point this out. METTL3 interacts with WTAP through its N-terminal (1-33aa) (1). Consistently, we find that the two cleaved forms METTL3a and METTL3b which lack the N-terminal region are not able to bind with WTAP. In Figure 4A, we overexpressed METTL3 WT as well as its different variants METTL3-Δ238, METTL3-Δ198, METTL3b and METTL3a respectively in WT cells, and compared the binding ability with WTAP or METTL14 across these overexpressed METTL3 variants. We acknowledge that the exogenous METTL3a and METTL3b interact with endogenous full-length METTL3, and the endogenous full-length METTL3 may help them to form the complex with WTAP. But in fact, the exogenous expression levels of METTL3a and METTL3b are much higher than that of endogenous full-length METTL3 (see Figure 3A and 3C). In this case, METTL3a or METTL3b predominantly interacts with itself, METTL3, METTL14 or other potential interacting proteins through its C-terminal region, this may greatly dilute the condition for the interaction between WTAP and endogenous full-length METTL3. Moreover, in Figure 4A, the comparison is among overexpressed METTL3 variants, the week indirect interactions through much lower expression levels of endogenous protein are probably not comparable to those direct interactions between overexpressed METTL3 variants and WTAP.

Reference:

1. Schöller, E., Weichmann, F., Treiber, T., Ringle, S., Treiber, N., Flatley, A., Feederle, R., Bruckmann, A., and Meister, G. (2018). Interactions, localization, and phosphorylation of the m6A generating METTL3–METTL14–WTAP complex. Rna 24, 499-512

**Reviewer #1 (Recommendations For The Authors):**
Major points:1. It is widely known that METTL3 and METTL14 form a stable heterodimer with the stoichiometric ratio of 1:1 (Wang X et al. Nature 534, 575-578 (2016), Su S et al. Cell Res 32(11), 982-994 (2022), Yan X et al. Cell Res 32(12), 1124-1127 (2022)), the numbers of METTL3 and METTL14 in the model of Fig 7P are not equivalent and need to be modified.

We thank for reviewer’s good suggestion. We have modified the model in Fig. 7P.

1. The in vitro methylation activity was detected by the m6A antibody, which has limited linear range. The MTase-Glo{trade mark, serif} Methyltransferase Assay is a SAMdependent enzyme assay with wide applications (Please refer to the references below).Could this assay be performed by authors?Wilkinson AW et al. Nature 565(7739), 372-376 (2019).Yu D et al. Nucleic Acids Res 49(20),11629-11642 (2021).Yan X et al. Cell Res 32(12), 1124-1127 (2022).Chen J et al. Nat Commun 13(1), 3257 (2022).

Thanks for reviewer’s good suggestion. We had performed the in vitro methylation assay by using MTase-Glo kit, and the data is consistent with the dot blot results. Please see the new Figure 2H-J.

1. When expressed alone in mammalian cell lines, METTL14 is unstable and is easily contaminated with endogenous METTL3 during purification (Yang W et al. Nat Cell Biol 16(2), p.191-8 (2014), Fig 1e). In Fig 2I, Co-expressing METTL3 and METTL14 maybe a good choice.

We thank for reviewer’s good suggestion. In fact, we co-expressed METTL3 and METTL14 in this in vitro methylation assay in Fig 2I (new Figure 2J in the revised version), METTL3-Flag or its mutant with Flag tag and METTL14-Flag were co-transfected into 293T cells, and co-purified by using Flag M2 magnetic beads from the cell lysates. We have added these details in the indicated method section, please see Line 574-585.

Other minor points:1. In Fig 5D, the protein domain information of METTL3 and relevant references need to be added (Su S et al. Cell Res 32(11), 982-994 (2022), Fig 6g; Yan X et al. Cell Res 32(12), 1124-1127 (2022), Fig 1a).

We have added these references in the revised manuscript.

1. In Fig 5, would METTL3b contribute to the METTL3-METTL3 interaction?

Our data showed that METTL3a but not METTL3b is responsible for the METTL3-WTAP interaction, breast cancer cell proliferation and the m6A modification. Then, we investigated the mechanism of how METTL3a regulates the METTL3-WTAP interaction, and found that METTL3a is essential for METTL3-METTL3 interaction, which is a prerequisite step for WTAP recruitment in MTC complex. In this case, we speculate that METTL3b is not required for the METTL3-METTL3 interaction. Indeed, through Co-IP assays,we found that METTL3b has no effect on the METTL3-METTL3 interaction (new supplementary Figure 4D), which is consistent with our above data showing that METTL3b is dispensable for the METTL3-WTAP interaction. We have added this comment in Page 6, Line 226 to 228.

1. In Fig 3F, the color in the legend and figure is inconsistent.

We have corrected the inconsistent color in the revised manuscript.

**Reviewer #2 (Recommendations For The Authors):**
1. In Figure 5D, the construction details of METTL3-HA and Flag should have been included in the method section. Are these tag sequences in the N-terminal of METTL3 protein?

These tags are all in the C-terminal of METTL3. We have added the construction details of these plasmids in the method section. Please see Line 434.

1. In Figure 7A, the labels of the inhibitors are overlapped with the figures.

We have corrected the labels of the inhibitors in Figure 7A in the revised manuscript.